# Vascular endothelial-specific loss of TGF-beta signaling as a model for choroidal neovascularization and central nervous system vascular inflammation

Yanshu Wang[1,2], Amir Rattner[1,2], Zhongming Li[1,2], Philip M Smallwood[1,2], Jeremy Nathans[1,2,3,4]*

[1]Department of Molecular Biology and Genetics, Johns Hopkins University School of Medicine, Baltimore, United States; [2]Howard Hughes Medical Institute, Johns Hopkins University School of Medicine, Baltimore, United States; [3]Department of Ophthalmology, Johns Hopkins University School of Medicine, Baltimore, United States; [4]Department of Neuroscience, Johns Hopkins University School of Medicine, Baltimore, United States

*For correspondence:
jnathans@jhmi.edu

Competing interest: The authors declare that no competing interests exist.

## eLife Assessment

Endothelial cell-specific loss of TGF-beta signaling in mice leads to CNS vascular defects, specifically impairing retinal development and promoting immune cell infiltration. The data are **solid**, showing that loss of TGF-beta signaling triggers vascular inflammation and attracts immune cells specific to CNS vasculature. These findings are **important**, highlighting TGF-beta's role in maintaining vascular-immune homeostasis and its therapeutic potential in neurovascular inflammatory diseases.

**Abstract** In mice, postnatal endothelial cell (EC)-specific knockout of the genes coding for transforming growth factor-beta receptor (TGFBR)1 and/or TGFBR2 eliminates TGF-beta signaling in vascular ECs and leads to distinctive central nervous system (CNS) vascular phenotypes. Knockout mice exhibit (1) reduced intraretinal vascularization, (2) choroidal neovascularization with occasional anastomoses connecting choroidal and intraretinal vasculatures, (3) infiltration of diverse immune cells into the retina, including macrophages, T-cells, B-cells, NK cells, and dendritic cells, (4) a close physical association between immune cells and retinal vasculature, (5) a pro-inflammatory transcriptional state in CNS ECs, with increased ICAM1 immunoreactivity, and (6) increased smooth muscle actin immunostaining in CNS pericytes. Comparisons of the retinal phenotype with two other genetic models of retinal hypovascularization – loss of Norrin/Fzd4 signaling and loss of vascular endothelial growth factor (VEGF) signaling – show that the immune cell infiltrate is greatest with loss of TGF-beta signaling, more modest with loss of Norrin/Fzd4 signaling, and undetectable with loss of VEGF signaling. The phenotypes caused by loss of TGF-beta signaling in ECs recapitulate some of the cardinal features of retinal and neurologic diseases associated with vascular inflammation. These observations suggest that therapies that promote TGF-beta-dependent anti-inflammatory responses in ECs could represent a promising strategy for disease modulation.

## Introduction

In the central nervous system (CNS), vascular development, structure, and function are tightly regulated by multiple signaling pathways, including the vascular endothelial growth factor (VEGF), Wnt, Notch, and TGF-beta pathways (*Wälchli et al., 2015*; *Wälchli et al., 2023*; *Rattner et al., 2022*). These signals control angiogenesis, vascular remodeling, and vascular permeability. With respect to permeability, the vasculature serving most of the CNS is distinguished from non-CNS vasculature by greatly reduced permeability, a specialization referred to as the blood-brain barrier (BBB) or, in the retina, the blood-retina barrier (BRB).

Disorders of CNS vascular structure and function are a major cause of morbidity and mortality (*Ropper et al., 2023*). These include arteriovenous malformations, vasculitis, atherosclerosis, small vessel disease, moyamoya, neovascular age-related macular degeneration (AMD), and diabetic retinopathy. Additionally, many CNS diseases or disorders that are not primarily vascular – including Alzheimer disease, multiple sclerosis, epilepsy, stroke, infection, and traumatic brain injury – are associated with altered vascular function, most commonly with reduced BBB integrity (*Profaci et al., 2020*; *Chen et al., 2024*).

A common feature of many CNS disorders is neuroinflammation. Although vascular endothelial cells (ECs) are not generally considered to be part of the immune system, in some locations and under some conditions, they acquire properties characteristic of immune cells, including secretion of cytokines, surface display of co-stimulatory or co-inhibitory receptors, and antigen presentation in association with MHC class II proteins (*Pober and Sessa, 2014*; *Amersfoort et al., 2022*). The best-characterized role for ECs in immune system function is as a site for binding and extravasation of circulating immune cells. As first demonstrated in lymphoid organs, ECs can recruit immune cells by expressing chemokines and cell-surface adhesion proteins that interact with cognate receptors on immune cells (*Sackstein, 2005*; *Blanchard and Girard, 2021*). In the context of inflammation, increased production of chemokines and adhesion proteins by ECs in non-lymphoid organs leads to the local recruitment of circulating immune cells and their subsequent extravasation (*Denes et al., 2024*). Under noninflammatory conditions, ECs are maintained in a quiescent state. In CNS ECs, quiescence is maintained in part by the actions of astrocyte-derived Sonic Hedgehog and endothelial Wnt signaling, with the result that few immune cells other than resident microglia are found within the CNS (*Alvarez et al., 2011*; *Lengfeld et al., 2017*).

TGF-beta signaling regulates inflammation in many tissues and in many cell types, as determined by the phenotypes of ligand, receptor, and downstream effector (SMAD) knockout (KO) mice (*Shull et al., 1992*; *Kulkarni et al., 1993*; *Travis and Sheppard, 2014*; *Massagué and Sheppard, 2023*). TGF-beta signaling also controls vascular development, with loss-of-function mutations in TGF-beta receptor 1 (TGFBR1/ALK5), TGFBR2, the EC accessory receptor Endoglin, or various SMADs producing defects in endothelial and/or pericyte development and early lethality (*Goumans et al., 2009*). The principal ligands for the TGFBR1-TGFBR2 heterodimer are the three highly homologous TGF-beta family members (*Heldin and Moustakas, 2016*). TGF-beta ligands are initially secreted as inactive ('latent') complexes that include inhibitory subunits, and the dimeric ligands are subsequently released by the catalytic action of integrins (*Shi et al., 2011*; *Dong et al., 2017*). Constitutive loss of both TGF-beta1 and TGF-beta3, loss of integrins expressed by glia, or loss of TGFBR2 in ECs produces similar defects in CNS angiogenesis and CNS vascular integrity (*Cambier et al., 2005*; *Mu et al., 2008*; *Aluwihare et al., 2009*; *Nguyen et al., 2011*; *Allinson et al., 2012*; *Arnold et al., 2012*; *Arnold et al., 2014*).

The role of TGF-beta signaling in retinal vascular development has been studied with CreER/LoxP conditional KO mice, both to facilitate survival beyond the early lethality of constitutive KO alleles and to permit the study of cell-type-specific KOs. Postnatal loss of TGFBR2 in all ocular cell types leads to microaneurysms, leaky capillaries, retinal hemorrhages, reactive microglia, and pericyte abnormalities, a picture that closely resembles the pathologies associated with severe diabetic retinopathy (*Braunger et al., 2015*). Postnatal EC-specific loss of TGFBR2 leads to defective retinal angiogenesis, the absence of an intraretinal capillary plexus, and choroidal neovascularization (CNV) (*Allinson et al., 2012*, *Schlecht et al., 2017*; *Zarkada et al., 2021*). The present study was undertaken to more fully define the cellular and molecular defects associated with endothelial-specific loss of TGF-beta signaling in the CNS, with an emphasis on the retina. In particular, we have sought to define the relationship between endothelial TGF-beta signaling and inflammation, as inflammation is a likely driver of diabetic retinopathy and AMD (*Wang et al., 2024*).

# Results

## CNV with loss of endothelial TGF-beta signaling

Retinal ECs were visualized with GS-lectin, anti-PECAM1, or anti-CLDN5 in retinas from young adult mice with early postnatal EC-specific loss of *Tgfbr1* or *Tgfbr2* (*Cdh5CreER;Tgfbr1^CKO/-^* or *Cdh5CreER;Tgfbr2^CKO/-^* mice treated with 4-hydroxytamoxifen [4HT] between postnatal day [P]3 and P5). These retinas show moderate disorganization of the three nuclear layers and a near absence of capillaries in the outermost tier of the retinal vasculature in the outer plexiform layer (OPL) (*Figure 1*; *Figure 1—figure supplement 1*), consistent with earlier descriptions of EC-specific inactivation of *Tgfbr1* (*Allinson et al., 2012*, *Schlecht et al., 2017*; *Zarkada et al., 2021*). These retinas also show multiple regions of choroidal (i.e. subretinal) neovascularization (CNV) as highlighted by the white arrows in *Figure 1A and B* (middle row) (see also *Figure 1C*, right, and *Figure 1—figure supplement 1B*).

Flatmounts of *Cdh5CreER;Tgfbr1^CKO/-^* retinas mounted with the photoreceptor side up and imaged at the level of the photoreceptors and retinal pigment epithelium (RPE) show many regions with scattered CD45+ immune cells (*Figure 1—figure supplement 1D*, upper panel) and zones of CNV that can be visualized in their entirety by immunostaining for a vascular marker (COL4 in *Figure 1—figure supplement 1D*, lower panel). Flatmounts of control retinas immunostained and imaged in the same manner show no CD45+ cells in the outer retina and no CNV. Subretinal zones with normal morphology or with CNV are also seen in toluidine blue-stained epon-embedded sections (*Figure 1—figure supplement 2*). Quantification of whole eye cross-sections shows an average of one to four zones of CNV per section in *Cdh5CreER;Tgfbr1^CKO/-^* and *Cdh5CreER;Tgfbr2^CKO/-^* retinas, but no detectable CNV in phenotypically WT control retinas (typically *Cdh5CreER;Tgfbr1^CKO/+^* or *Cdh5CreER;Tgfbr2^CKO/+^*) or in retinas with severely reduced intraretinal vascular development secondary to loss of Norrin/Fzd4 signaling (*Ndp^KO^*; *Figure 1—figure supplement 1C*). Although the quantification in *Figure 1—figure supplement 1C* shows more CNV zones per section in *Cdh5CreER;Tgfbr2^CKO/-^* retinas compared to *Cdh5CreER;Tgfbr1^CKO/-^* retinas, we think it likely that a more extensive sampling would show little or no difference between these two genotypes.

Throughout this study, our analyses of EC-specific KO of *Tgfbr1* (*Cdh5CreER;TgfbrR1^CKO/-^*), *Tgfbr2* (*Cdh5CreER;TgfbrR2^CKO/-^*), and of both *Tgfbr1* and *Tgfbr2* (*Cdh5CreER;TgfbrR1^CKO/-^;Tgfbr2^CKO/-^*) indicate that the vascular phenotypes in the brain and retina are virtually identical across all three genotypes, consistent with current models that envision TGFBR1-TGFBR2 heterodimers as the active receptor complex (e.g. compare *Figure 1A and B* and compare *Figure 1—figure supplements 1 and 3*). In the figures that follow, all analyses have been conducted with EC-specific KO of *Tgfbr1*. For completeness, we also include some examples from EC-specific KO of *Tgfbr2* or EC-specific double KO of *Tgfbr1* and *Tgfbr2*.

In wild-type (WT) mice, all retinal ECs are PECAM1+ (platelet and endothelial cell adhesion molecule-1) and CLDN5+ (Claudin5, a marker of BBB and BRB vasculature) and all of the choroidal ECs are PLVAP+ (plasmalemma vesicle-associated protein, a marker of permeable vasculature) (*Figure 1B and C*; *Figure 1—figure supplement 3*). In mice with EC-specific deletion of *Tgfbr1* or *Tgfbr2*, all of the choroidal ECs are PLVAP+, all of the retinal ECs remain PLVAP-, and most, but not all, retinal ECs are CLDN5+ (*Figure 1B and C*; *Figure 1—figure supplement 3*). We note that this PLVAP phenotype contrasts with the phenotype caused by mutations in the Norrin/Fzd4 pathway, which converts all retinal ECs to PLVAP+ and all retinal capillary and vein ECs to CLDN5- (*Wang et al., 2018*).

Mutant retinas also exhibit occasional anastomoses between the subretinal and retinal vasculatures, a feature never observed in control retinas (*Figure 1B*, white arrows in bottom panels). The intraretinal segments of the communicating vessels are CLDN5+/PLVAP- and the subretinal segments, which invariably arise in a zone of CNV, are CLDN5-/PLVAP+ (*Figure 1B*), implying a mixed origin for these vessels. The attenuated intraretinal vasculature produced by deficient TGF-beta signaling is associated with retinal hypoxia as determined by localized accumulation of hypoxia-inducible factor (HIF)1-alpha in the nuclei of retinal parenchymal cells (*Figure 1—figure supplement 4*).

In mutant retinas, CD45+ immune cells are abundant, and many of these cells are closely associated with the subretinal and inner retinal vasculatures (*Figure 1A and C*; *Figure 1—figure supplement 1A and B*). In control retinas, the only CD45+ cells are the relatively sparse microglia. In the several non-CNS tissues examined – heart, kidney, liver, and lung – the density of CD45+ cells appears to be unaffected in *Cdh5CreER;Tgfbr1^CKO/-^* mice (*Figure 1—figure supplement 5*). Immunostaining for RPE65, a marker for the RPE, and toluidine blue-staining of epon-embedded retina sections both

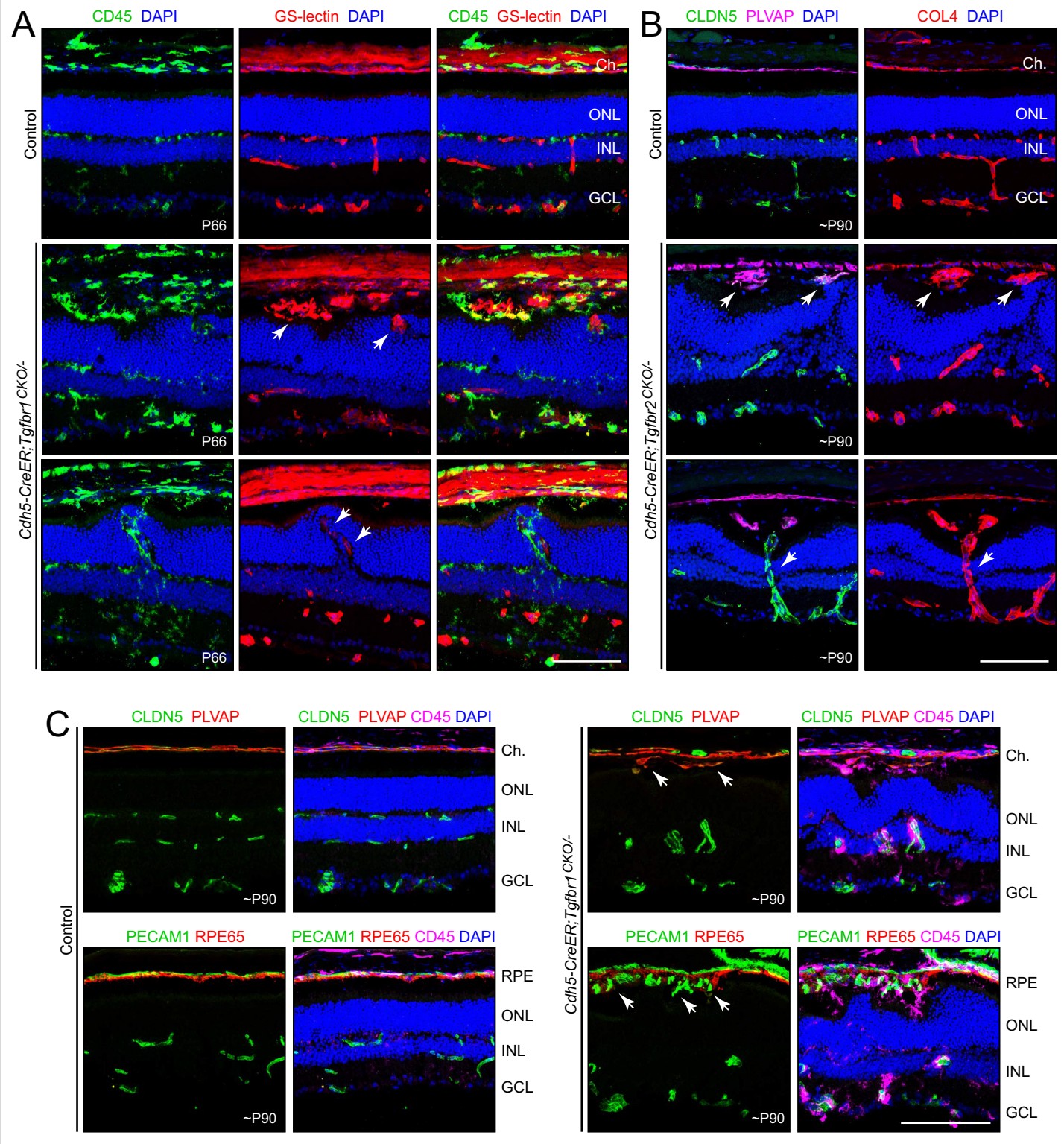

**Figure 1.** Endothelial cell (EC)-specific loss of TGF-beta signaling leads to attenuated retinal vascular development, choroidal neovascularization (CNV), and anastomoses between retinal and choroidal vasculatures. (**A**) *Cdh5CreER;Tgfbr1$^{CKO/-}$* retinas showing CNV (white arrows in central panels) and vascular invasion of the outer nuclear layer (white arrows in lower panels), both with associated CD45+ immune cells. (**B**) *Cdh5CreER;Tgfbr2$^{CKO/-}$* retinas showing CNV (white arrows in central panels) and an anastomosis between retinal and choroidal vasculatures (white arrows in lower panels), with intraretinal EC marker CLDN5, choroidal EC marker PLVAP, and pan-vascular marker COL4 (collagen4). (**C**) Upper right panels, in the *Cdh5CreER;Tgfbr1$^{CKO/-}$* retina, CNV (white arrows) is derived from choroidal vasculature, marked by PLVAP. Lower right panels, CNV (white arrows) is

*Figure 1 continued on next page*

*Figure 1 continued*

present in the subretinal space, i.e., on the retinal side of the RPE, which is marked by RPE65. Abbreviations: Ch., choroid; ONL, outer nuclear layer; INL, inner nuclear layer; GCL, ganglion cell layer; RPE, retinal pigment epithelium. The ages of the mice are indicated in postnatal days (P) for this and all other figures. Scale bars, 100 µm.

The online version of this article includes the following figure supplement(s) for figure 1:

**Figure supplement 1.** Localized choroidal neovascularization (CNV) with endothelial cell (EC)-specific loss of TGF-beta signaling.

**Figure supplement 2.** Endothelial cell (EC)-specific loss of TGF-beta signaling leads to defects in subretinal structure, with alternating zones of normal structure and choroidal neovascularization (CNV).

**Figure supplement 3.** Endothelial cell (EC)-specific loss of TGF-beta signaling produces little or no change in pericyte NG2 immunostaining.

**Figure supplement 4.** Retinal hypoxia in *Cdh5CreER;Tgfbr1CKO/-* retinas at postnatal day (P)18.

**Figure supplement 5.** No change in the density of CD45+ immune cells in *Cdh5CreER;Tgfbr1CKO/-* heart, kidney, liver, and lung at postnatal day (P)14.

**Figure supplement 6.** Endothelial cell (EC) specificity of *Cdh5CreER* assessed by recombination of two *loxP-stop-loxP* (*LSL*) reporters.

**Figure supplement 7.** Endothelial cell (EC) specificity of *Cdh5CreER* compared to CD45+ immune cells, assessed by recombination of the *Rosa26-LSL-SUN1-sfGFP-6xmyc* reporter.

**Figure supplement 8.** Endothelial cell (EC) specificity of *Cdh5CreER* in the *Cdh5CreER;Tgfbr1CKO/-* genetic background, based on recombination with the *Rosa26-LSL-SUN1-sfGFP-6xmyc* reporter.

show RPE displacement at the sites of CNV (*Figure 1C*; *Figure 1—figure supplements 1B and 2*). In mutant retinas, there is little change in pericyte NG2 immunostaining (*Figure 1—figure supplement 3*).

As the phenotypes associated with EC-specific deletion of *Tgfbr1* or *Tgfbr2* involve both ECs and immune cells, it is important to ask whether Cre-mediated recombination directed by *Cdh5CreER* is, in fact, EC-specific. With two *LoxP-stop-LoxP* (*LSL*) reporters that express nuclear-localized GFP, *Rosa26-LSL-mtdT-2A-nlsGFP* (*Wang et al., 2018*) and *Rosa26-LSL-SUN1-sfGFP-6xmyc* (*Mo et al., 2015*), *Cdh5CreER* directs recombination exclusively in ECs in the retina, cerebellum, heart, kidney, and lung, as judged by cellular morphology, EC identification with PECAM1 immunostaining, and co-localization of GFP with the EC-specific transcription factor ERG (*Figure 1—figure supplement 6*). Additionally, CD45 localization in immune cells is mutually exclusive with *Rosa26-LSL-SUN1-sfGFP-6xmyc* reporter expression in the choroid, small intestine, and retina (*Figure 1—figure supplement 7*). In these experiments, the occasional exceptions to ERG and GFP co-localization consist of ECs in which Cre-mediated recombination failed to occur (e.g. *Figure 1—figure supplement 6B*, white arrows in the kidney and lung panels). As a further test of *Cdh5CreER* specificity, we asked whether the EC-specific recombination of the *Rosa26-LSL-SUN1-sfGFP-6xmyc* reporter observed on a WT *Tgfbr1* background was retained in a *Tgfbr1CKO/-* background. As seen in *Figure 1—figure supplement 8*, the CD45+ immune cells that accumulate in the retina lacking endothelial *Tgfbr1* do not express the GFP reporter, which retains its pattern of EC specificity. Based on these experiments, we conclude that all of the phenotypes observed in *Cdh5CreER;Tgfbr1CKO/-* and *Cdh5CreER;Tgfbr2CKO/-* mice follow from the loss of TGFBR function exclusively within ECs, and, more specifically, that all immune cell phenotypes in the retina are secondary to a change in EC properties.

*Cdh5CreER;Tgfbr1CKO/-* retina flatmounts show large numbers of vascular tufts (*Figure 2A*). This aberrant vascular architecture, which is seen in both *Cdh5CreER;Tgfbr1CKO/-* and *Cdh5CreER;Tgfbr2CKO/-* mice, shows some similarities to the previously reported vascular architecture of *Fzd4-/-* and *NdpKO* mice (*Luhmann et al., 2005*; *Ye et al., 2009*). In particular, retinas with each of these genotypes variably display disorganized and hypertrophic superficial vessels that give rise to intraretinal vascular tufts instead of the well-organized trilayered retinal vasculature. The morphologies and locations of the intraretinal vascular tufts are similar among all of these mutant genotypes, as shown in *Figure 2B*. At P14, when endothelial tip cells are building the central tier of intraretinal capillaries in WT (control) retinas (white arrows in the upper left panel in *Figure 2B*), ECs in *Cdh5CreER;Tgfbr1CKO/-* retinas have formed only a few rudimentary capillaries or disconnected vascular tufts (lower left panels in *Figure 2B*).

Vascular permeability to the low-molecular-weight amine-reactive intravascular tracer Sulfo-NHS-biotin (administered by intraperitoneal [IP] injection ~30 min before sacrifice) was compared among control, *Cdh5CreER;Tgfbr1CKO/-*, and *NdpKO* retinas at P14 (*Figure 2C*). Control retinas show little Sulfo-NHS-biotin accumulation in the vasculature or in the parenchyma (i.e. the area between the

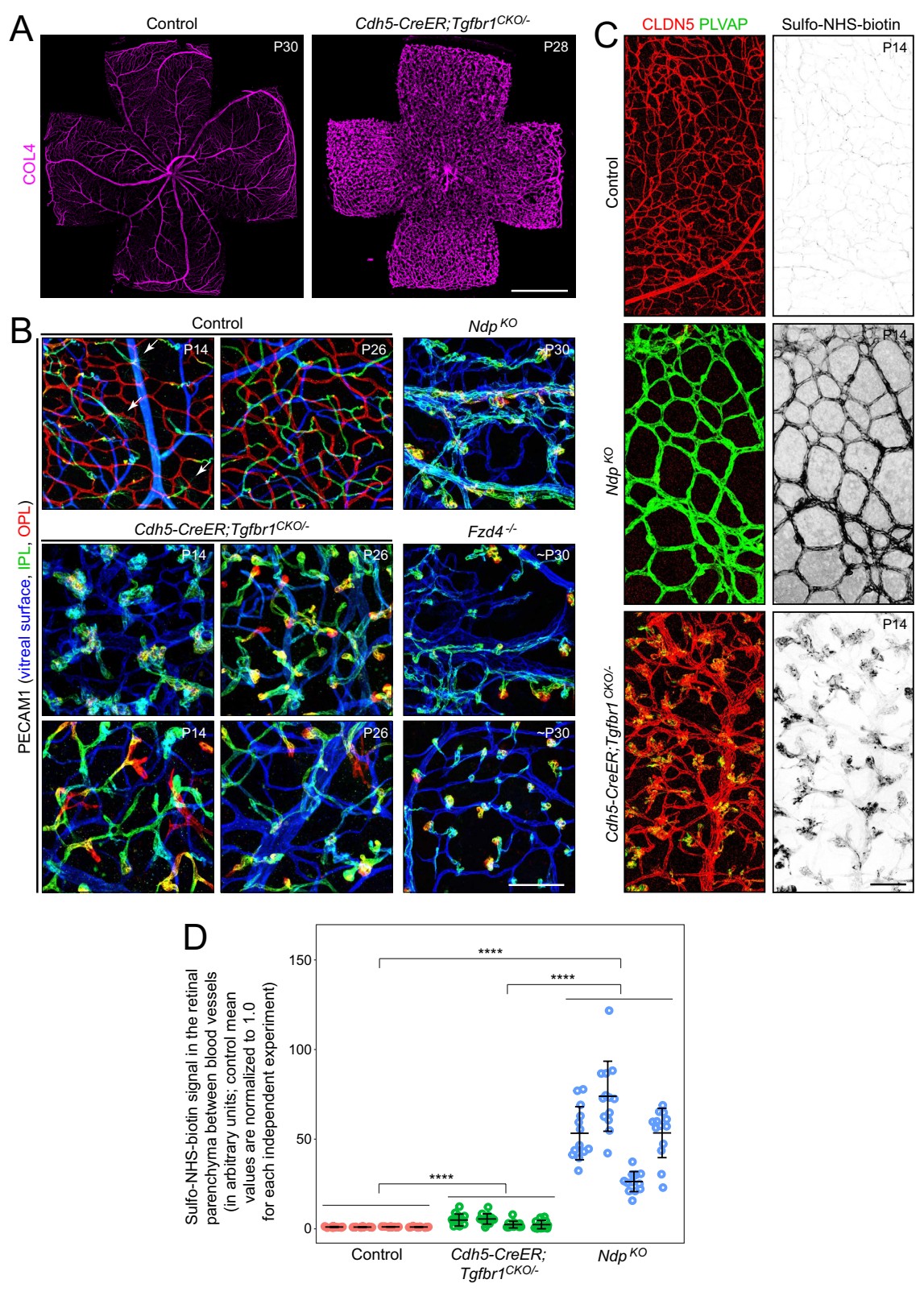

**Figure 2.** Vascular architecture in retinas with endothelial cell (EC)-specific loss of TGF-beta signaling or global loss of Norrin/Fzd4 signaling. (**A**) COL4 immunostaining of control and *Cdh5CreER;Tgfbr1^CKO/-* flatmount retinas showing arteries, veins, and capillaries in the control retina and a high density of vascular tufts in the *Cdh5CreER;Tgfbr1^CKO/-* retina. (**B**) False color images from the indicated genotypes and ages showing a stacked Z-series of flatmount retinas color-coded by the depth of the vasculature. For control retinas, the blue-green-red color scheme corresponds to the inner two-thirds

*Figure 2 continued on next page*

*Figure 2 continued*

of the retina: blue, vitreal surface; green, ganglion cell layer and inner plexiform layer; and red, inner nuclear layer and outer plexiform layer. For mutant retinas, the blue-green-red color scheme corresponds to a shallower depth, as the most deeply penetrating vascular tufts go only as far as the inner edge of the inner nuclear layer: blue, vitreal surface; green, ganglion cell layer; and red, inner plexiform layer. Left column of three panels: postnatal day (P)14 control retina (upper image; white arrows point to tip cells in the inner plexiform layer [IPL]) and two regions from a P14 *Cdh5CreER;Tgfbr1^CKO/-* retina (lower). Center column of three panels: P26 control retina (upper) and two regions from a P26 *Cdh5CreER;Tgfbr1^CKO/-* retina (lower). Right column of three panels, ~P30 *Ndp^KO* retina (upper) and two regions from an ~P30 *Fzd4^-/-* retina (lower). All images are at the same magnification and are from the midperiphery of the retina. (**C**) Flatmounts of P14 control, *Ndp^KO*, and *Cdh5CreER;Tgfbr1^CKO/-* retinas showing CLDN5 and PLVAP immunostaining and Sulfo-NHS-biotin accumulation (detected with fluorescent streptavidin). (**D**) Quantification of Sulfo-NHS-biotin in the retinal parenchyma. Pixel intensities for territories between vascular segments were quantified from four flatmount retinas for each of the three genotypes (13 territories per retina). For the two cohorts of control and mutant retinas, all pixel intensity values were scaled so that the mean values from the control retinas equal 1.0. Scale bar in (**A**), 1 mm. Scale bar in (**B**), 100 µm. Scale bar in (**C**), 100 µm.

vessels); *Ndp^KO* retinas show Sulfo-NHS-biotin accumulation in the vasculature and in the parenchyma, and *Cdh5CreER;Tgfbr1^CKO/-* retinas show Sulfo-NHS-biotin accumulation in the vascular tufts with minimal accumulation in the non-tuft regions of vasculature and minimal leakage into the parenchyma (*Figure 2C and D*). As reported previously and as seen in *Figure 2C*, control retinal ECs are CLDN5+/PLVAP- and *Ndp^KO* retinal ECs are CLDN5-/PLVAP+ (*Wang et al., 2018*). In *Cdh5CreER;Tgfbr1^CKO/-* retinas, ECs in the bulk of the vasculature are CLDN5+/PLVAP-, but many ECs in the vascular tufts also express low levels of PLVAP. In these retinas, Sulfo-NHS-biotin accumulation in the tufts may reflect a local increase in vascular permeability. It is also worth noting that blood flow within the vascular tufts is almost certainly aberrant, as these structures comprise dead ends rather than intermediary elements in the capillary network. This anomalous blood flow could also affect Sulfo-NHS-biotin accumulation.

## Immune cell phenotype in the retina with loss of endothelial TGF-beta signaling

To more rigorously characterize the immune response in *Cdh5CreER;Tgfbr1^CKO/-* retinas and to compare this response to that observed with loss of Norrin/Fzd4 signaling (*Ndp^KO* retinas and *Fzd4^-/-* retinas), flatmount retinas from P15 to P30 mice were immunostained for CD45/PTPRC (multiple immune cell types), F4-80 (monocytes and macrophages), PU.1/SPI1 (myeloid cells), IBA1/AIF-1 (microglia and macrophages), and CD3E (T-cells) (*Figure 3A–C*). Each of these markers shows an increase in immune cells in *Cdh5CreER;Tgfbr1^CKO/-* retinas compared to age-matched controls. Quantifying the density of cells staining for CD45, PU.1, F4-80, and CD3E shows that, for each of these markers, endothelial loss of TGF-beta signaling is associated with the highest immune cell increase and loss of Norrin/Fzd4 signaling is associated with a more modest immune cell increase (*Figure 3D–G*). Interestingly, CreER-mediated recombination in *Cdh5CreER;Tgfbr1^CKO/-* mice after the completion of retinal vascular development (~P14) did not lead to immune cell infiltration in the retina (data not shown), suggesting that the abnormal development/anatomy of the *Cdh5CreER;Tgfbr1^CKO/-* retinal vasculature plays an essential role in the immune cell phenotype. This could reflect a role for retinal hypoxia in the pro-inflammatory state.

In the quantification of CD45+ cells in *Figure 3*, cells with resident microglial morphology were not counted. As seen by immunostaining for ASC (apoptosis-associated speck-like protein containing a CARD; nuclei) and CD45 (plasma membrane), microglia are present in control retina flatmounts at a density of 25–30 cells per 450 µm × 450 µm area in each of the three retinal layers in which they reside (retinal ganglion cell layer, inner plexiform layer, and OPL), for a total density of ~85 cells per 450 µm × 450 µm area (*Figure 3—figure supplement 1*). The density of CD45+ cells in *Fzd4^-/-* and *Ndp^KO* retinas in excess of the density of CD45+ cells in control retinas is ~100 per 450 µm × 450 µm area, and the excess density of CD45+ cells in *Cdh5CreER;Tgfbr1^CKO/-* retinas is ~200 per 450 µm × 450 µm area (*Figure 3D*).

In flatmounts of control choroids, CD45+ cells are present at a density of 100–150 cells per 645 µm × 645 µm area, and in flatmounts of *Cdh5CreER;Tgfbr1^CKO/-* choroid, CD45+ cells are present at approximately twice that density, with substantial scatter in the *Cdh5CreER;Tgfbr1^CKO/-* data (*Figure 3—figure supplement 2*). A curious feature of CD45+ cells in the *Cdh5CreER;Tgfbr1^CKO/-* retina is the large number of cells that are positive for cleaved caspase 3, a marker of apoptosis (*Figure 3—figure supplement 3*). Very few retinal cells of any other type are positive for cleaved

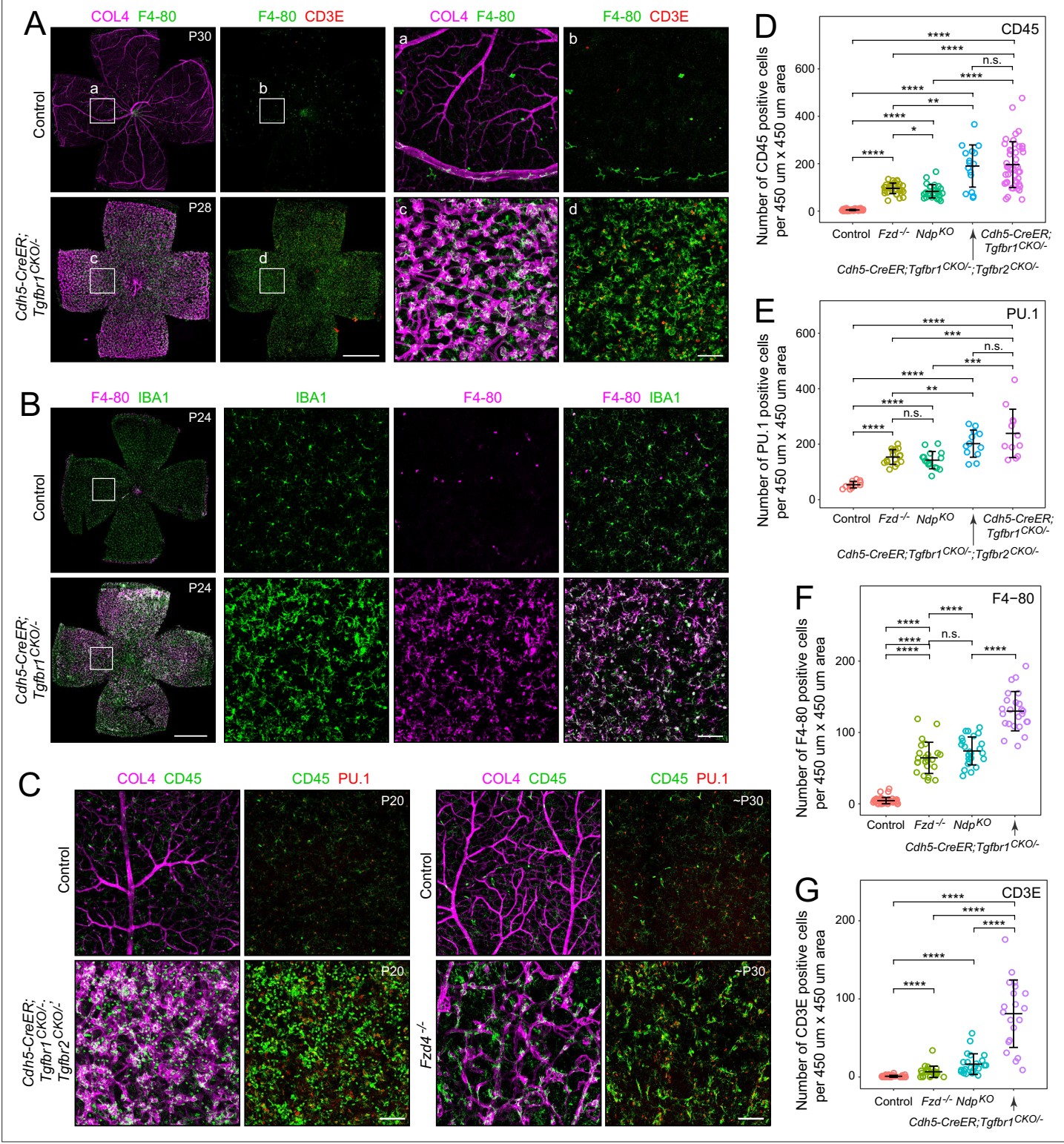

**Figure 3.** Immune cells in retinas with endothelial cell (EC)-specific loss of TGF-beta signaling or global loss of Norrin/Fzd4 signaling. (**A**) Control and *Cdh5CreER;Tgfbr1^CKO/-* retina flatmounts (left pair of panels) with enlarged insets (right pair of panels). (**B**) Control and *Cdh5CreER;Tgfbr1^CKO/-* retina flatmounts (left panel) and insets (right three panels). (**C**) Control and *Cdh5CreER;Tgfbr1^CKO/-;Tgfbr2^CKO/-* retina flatmounts, and control and *Fzd4^-/-* retina flatmounts. (**D–G**) Quantification of numbers of cells positive for the indicated markers in 450 μm × 450 μm zones in the midperiphery of retina flatmounts from mice of the indicated genotypes. In this and other panels with statistical comparisons, the bars represent mean ± standard deviation,

*Figure 3 continued on next page*

*Figure 3 continued*

and p-values, calculated using the Wilcoxon rank-sum test, are shown as *<0.05, **<0.01, ***<0.001, and ****<0.0001. Scale bars in whole retina panels, 1 mm. Scale bars in all other panels, 100 μm.

The online version of this article includes the following figure supplement(s) for figure 3:

**Figure supplement 1.** Density of microglia in control (wild-type) flatmount retinas.

**Figure supplement 2.** Increased macrophage density in the choroid with endothelial cell (EC)-specific loss of TGF-beta signaling.

**Figure supplement 3.** Large number of apoptotic immune cells in retinas with endothelial cell (EC)-specific loss of TGF-beta signaling.

caspase 3. In control retinas, and in $Fzd4^{-/-}$ and $Ndp^{KO}$ retinas, very few cells are positive for cleaved caspase 3, despite the presence of excess CD45+ cells in $Fzd4^{-/-}$ and $Ndp^{KO}$ retinas (*Figure 3—figure supplement 3A*; quantified for $Fzd4^{-/-}$ in *Figure 3—figure supplement 3B*). These data imply that in the $Cdh5CreER;Tgfbr1^{CKO/-}$ retina, there is rapid turnover of CD45+ cells, with new cells replenishing the population as resident cells are eliminated.

## Immune cell phenotyping in the retina by single-nucleus RNAseq

To obtain an unbiased assessment of the immune cell types present in the $Cdh5CreER;Tgfbr1^{CKO/-}$ retina, P14 control and $Cdh5CreER;Tgfbr1^{CKO/-}$ retinas were compared by single-nucleus (sn)RNAseq (*Figure 4*). For both genotypes, immune cell nuclei represent only a minor fraction of retinal nuclei (*Figure 4A*; N=628 immune cell nuclei). While microglia are present in both control and $Cdh5CreER;Tgfbr1^{CKO/-}$ retinas, all of the other immune cells in the combined snRNAseq dataset were derived from $Cdh5CreER;Tgfbr1^{CKO/-}$ retinas (*Figure 4B*). As determined by the patterns of expression of known immune cell markers (*BD Biosciences, 2024*), the immune cell population in $Cdh5CreER;Tgfbr1^{CKO/-}$ retinas encompasses the full spectrum of major cell classes: B-cells, T-cells, dendritic cells, macrophages, natural killer (NK)-cells, and microglia (*Figure 4C–E*). In contrast to the immune cells in the $Aire^{-/-}$ mouse model of autoimmune uveoretinitis (*Heng et al., 2019*), which self-organize into tertiary lymphoid organs, the immune cells in $Cdh5CreER;Tgfbr1^{CKO/-}$ retinas show little evidence of spatial organization beyond an association with the vasculature (*Figure 3A–C* and text below).

## Hypovascularization secondary to loss of retinal VEGF signaling does not lead to immune cell infiltration

The presence of immune cells in retinas lacking endothelial TGF-beta signaling or Norrin/Fzd4 signaling could potentially arise from immune cell infiltration secondary to the hypoxic stress within the inner retina that results from reduced vasculature (*Figure 1—figure supplement 4*). Alternately, it could reflect changes in the intrinsic properties of retinal ECs that promote immune cell recruitment and extravasation. As a first step in distinguishing between these models, we generated and studied a distinct retinal hypovascularization model – $Vsx2$-$Cre;Vegfa^{CKO/CKO}$ – in which deletion of VEGF-A (hereafter, VEGF) in retinal Müller glia and neurons (*Rowan and Cepko, 2004*) greatly reduces intra-retinal vascularization.

VEGF production by surface astrocytes guides the first stage of retinal angiogenesis, during which ECs grow outward from the optic disc along the vitreal face of the retina (*Stone et al., 1995*; *Rattner et al., 2019*). The second stage of retinal angiogenesis involves EC growth into the retina and is driven by VEGF production from cells within the inner retina (*Rattner et al., 2019*). In $Vsx2$-$Cre;Vegfa^{CKO/CKO}$ mice, the first stage of retinal angiogenesis is largely unaffected, but the second stage fails to occur (*Figure 5A–C*). In retinas lacking endothelial TGF-beta signaling or Norrin/Fzd4 signaling, high levels of retinal VEGF drive excess proliferation of ECs on the vitreal face of the retina and in intraretinal tufts. In contrast, the lack of retina-derived VEGF in $Vsx2$-$Cre;Vegfa^{CKO/CKO}$ mice results in little or no additional proliferation of ECs (*Figure 5A–C*).

To compare immune cell infiltration between control and $Vsx2$-$Cre;Vegfa^{CKO/CKO}$ retinas, flatmount retinas were immunostained for ASC and CD45 (*Figure 5D*). Both sets of retinas exhibited similar numbers of immune cells, almost all of which appear to be microglia as judged by their morphology and distribution (*Figure 5E*). These data imply that retinal hypo-vascularization and inner retinal hypoxia per se do not lead to immune cell infiltration into the retina. Instead, the simplest explanation for this data is that the absence of endothelial TGF-beta signaling – and, to a lesser extent, the absence of

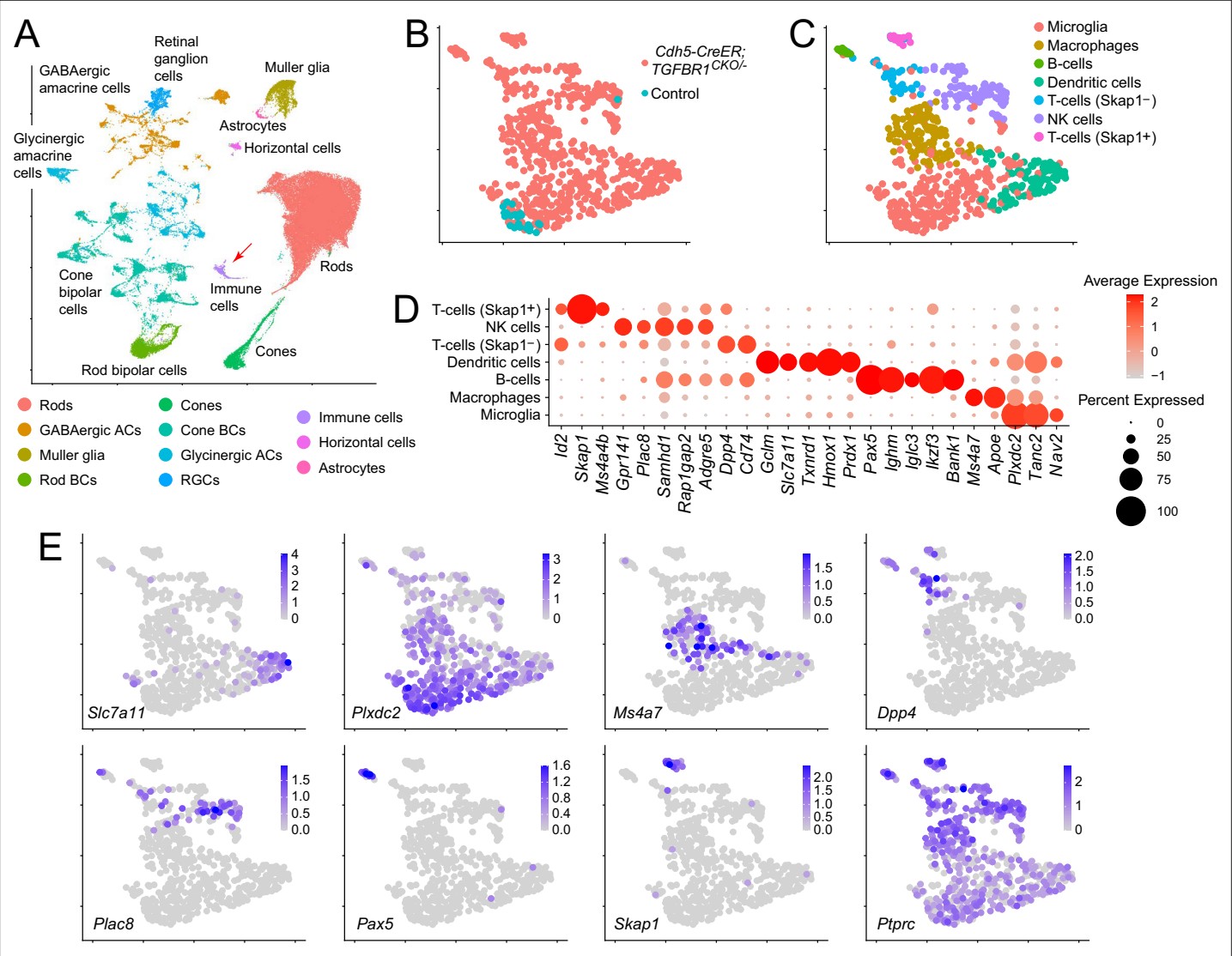

**Figure 4.** Single-nucleus RNA sequencing from control and *Cdh5CreER;Tgfbr1^{CKO/-}* retinas at postnatal day (P)14. (**A**) Identification of cell clusters in a Uniform Manifold Approximation and Projection (UMAP) plot, based on established markers of retinal gene expression. The pooled control and *Cdh5CreER;Tgfbr1^{CKO/-}* data are shown. The red arrow points to immune cells, almost entirely derived from the *Cdh5CreER;Tgfbr1^{CKO/-}* samples, as shown in (**B**). (**B**) The contributions of control and *Cdh5CreER;Tgfbr1^{CKO/-}* retinas to the immune cell cluster. (**C**) UMAP plot identifying immune cell types within the immune cell cluster, based on established markers as shown in (**D**) and (**E**). (**D**) Abundances of select transcripts in different immune cell types. (**E**) UMAP plots for select markers from (**D**) showing immune cell-type-specific expression.

Norrin/Fzd4 signaling, but not the absence of VEGF signaling – leads to a pro-inflammatory vascular state that attracts immune cells. This hypothesis is explored in the text and figures below.

## Close physical association between retinal vasculature and immune cells with loss of endothelial TGF-beta signaling

A close association between blood vessels and CD45+ immune cells was observed in the retina cross-sections in *Figure 1*. To visualize these associations in intact retinas, flatmounts were immunostained for PECAM1, ASC, and CD45 (*Figure 6*). In control retinas, resident microglia are the only CD45+/ASC+ cells in the retina and their cell bodies and processes tile the retina in a manner that appears to be independent of the locations of blood vessels (*Figure 6*, *Figure 3—figure supplement 1*, and *Figure 6—figure supplements 1–3*). By contrast, in retinas with loss of EC TGF-beta signaling, many immune cells are closely associated with the vasculature (*Figure 6*, *Figure 6—figure supplements 1–3*). Aberrant vessels that penetrate the full thickness of the outer retina are associated along their

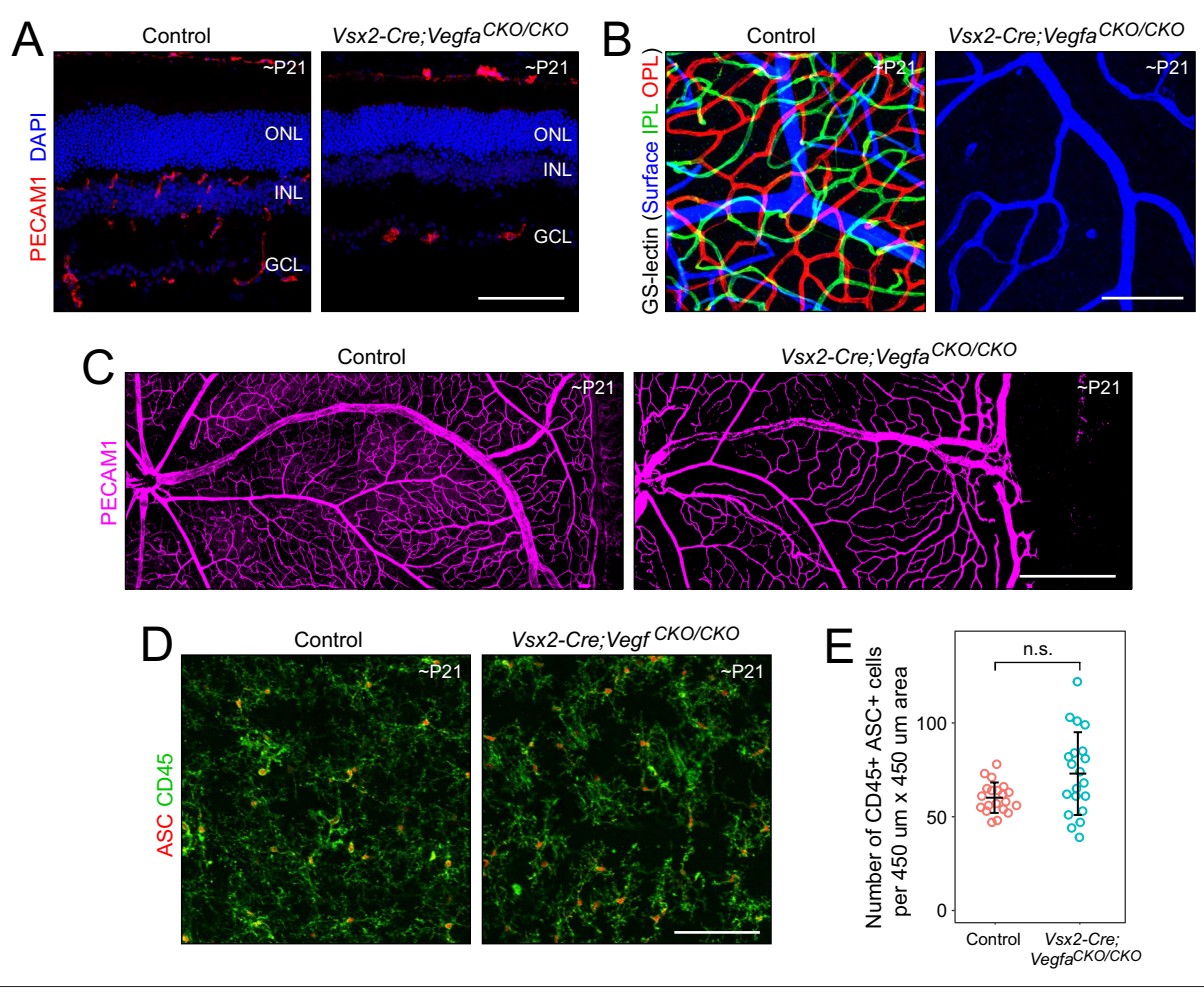

**Figure 5.** Vascular anatomy and absence of immune cell infiltration in *Vsx2-Cre;Vegfa*<sup>CKO/CKO</sup> retinas. (**A**) Sections of control and *Vsx2-Cre;Vegfa*<sup>CKO/CKO</sup> retinas immunostained with anti-PECAM1 to visualize the vasculature. (**B**) False color images of control and *Vsx2-Cre;Vegfa*<sup>CKO/CKO</sup> flatmount retinas showing a stacked Z-series color-coded by the depth of PECAM1 immunostained vasculature. Blue, vitreal surface; green, inner plexiform layer; red, outer plexiform layer. (**C**) Control and *Vsx2-Cre;Vegfa*<sup>CKO/CKO</sup> flatmount retinas immunostained with anti-PECAM1. (**D**) Control and *Vsx2-Cre;Vegfa*<sup>CKO/CKO</sup> flatmount retinas immunostained for ASC and CD45. (**E**) Quantification of CD45+/ASC+ cells. Scale bars, (**A**), (**B**), and (**D**), 100 μm; (**C**) 500 μm.

length with CD45+ cells, as seen in serial Z-plane images at both low magnification (*Figure 6—figure supplements 1 and 2*) and high magnification (*Figure 6—figure supplement 3*).

The subcellular distribution of PECAM1 in retinal vasculature is nonuniform in both mutant and control retinas (*Figure 6D*). Close inspection of *Cdh5CreER;Tgfbr1*<sup>CKO/-</sup> retinas shows that in venous ECs there are numerous ~5 μm diameter zones of reduced PECAM1 immunostaining, often accompanied by a rim of increased staining (white arrows in *Figure 6D*). These zones coincide with the locations of CD45+ immune cells, implying a direct physical association between immune cells and ECs (*Figure 6D*). Although one cannot determine, at the resolution of these images, whether the PECAM1 signal derives from ECs, immune cells, or both, these PECAM1+ rings likely represent the 'transmigratory cup' that has been described as part of the process of leukocyte diapedesis and which appears to involve homophilic PECAM1-PECAM1 binding between ECs and leukocytes (*Carman and Springer, 2004*; *Mamdouh et al., 2009*; *Arif et al., 2021*). This association is only rarely observed in control retinas, where immune cells other than microglia are sparse (*Figure 6D*). In *Cdh5CreER;Tgfbr1*<sup>CKO/-</sup> retinas, the association of leukocytes with veins is reminiscent of leukocyte adhesion to high endothelial venules in lymphoid organs (*Blanchard and Girard, 2021*). These data imply that ECs lacking TGF-beta signaling bind immune cells, likely promoting their recruitment and extravasation.

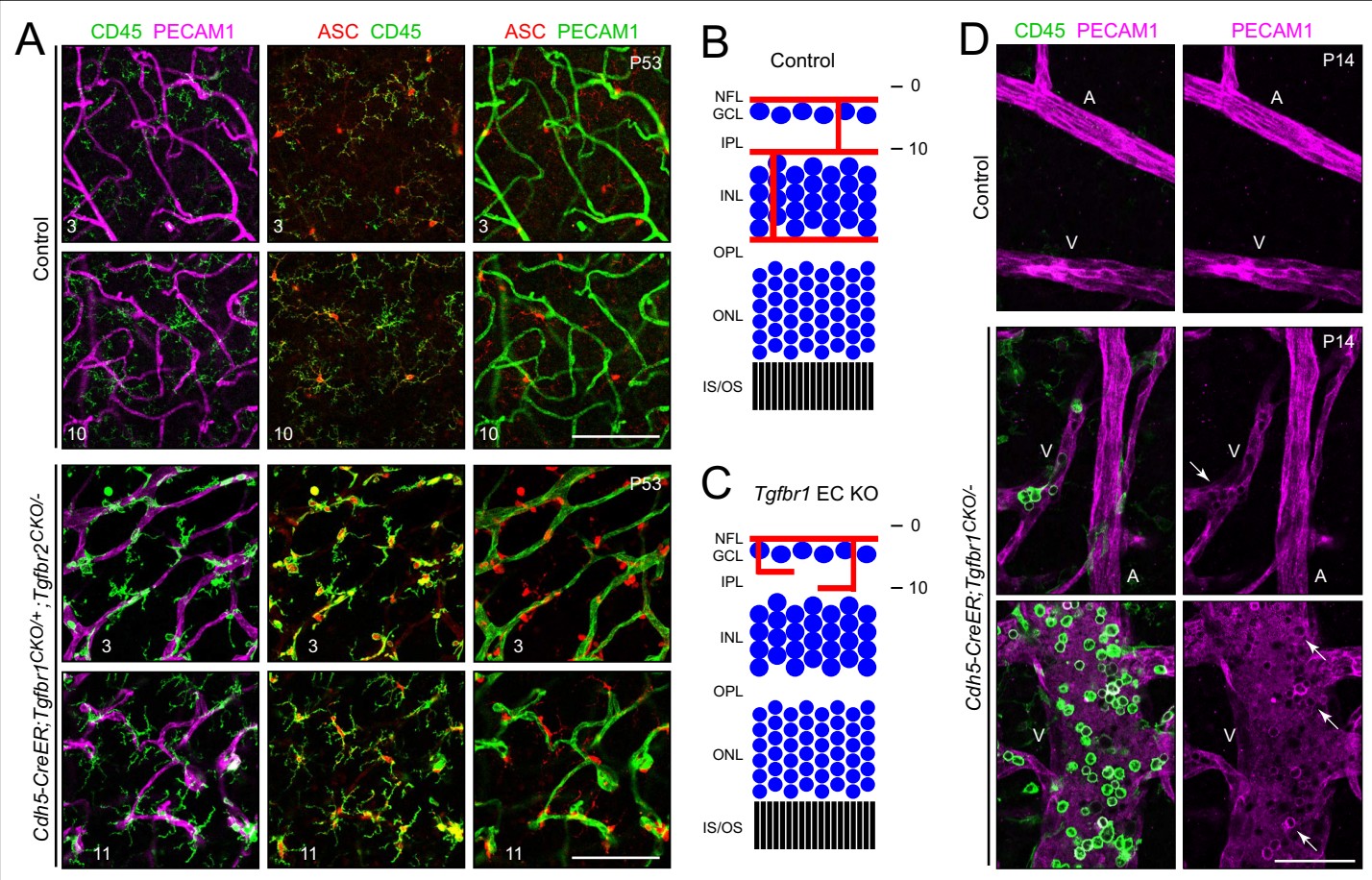

**Figure 6.** Close association between immune cells and retinal vasculature with endothelial cell (EC)-specific loss of TGF-beta signaling. (**A**) Upper six panels show a control retina flatmount. Lower six panels show a *Cdh5CreER;Tgfbr1^{CKO/+};Tgfbr2^{CKO/-}* retina flatmount. ASC and CD45 label immune cells, including microglia. The Z-plane is indicated by the numbers at the bottom of each image. The nerve fiber layer (Z-plane 3) and the inner plexiform layer (Z-planes 10–11) are shown schematically in (**B**) and (**C**). (**B** and **C**) Retinal schematics showing the relationship of the vasculature and the three retinal layers. Confocal Z-planes are numbered at right. (**D**) Immune cells and venous ECs in control and *Cdh5CreER;Tgfbr1^{CKO/-}* retinas. In the lower image, three of the 'impressions' of CD45+ immune in the distribution of PECAM1 on the EC surface are highlighted with white arrows. A, artery; V, vein. NFL, nerve fiber layer; GCL, ganglion cell layer; IPL, inner plexiform layer; INL, inner nuclear layer; OPL, outer plexiform layer; ONL, outer nuclear layer; IS/OS, inner segment/outer segment. Scale bars in (**A**), 100 μm. Scale bar in (**D**), 50 μm.

The online version of this article includes the following figure supplement(s) for figure 6:

**Figure supplement 1.** Close association between immune cells and a blood vessel forming a retinal-choroidal anastomosis in a retina with endothelial cell (EC)-specific loss of TGF-beta signaling.

**Figure supplement 2.** Low-magnification view of the close association between immune cells and a blood vessel forming a retinal-choroidal anastomosis in a retina with endothelial cell (EC)-specific loss of TGF-beta signaling.

**Figure supplement 3.** High-magnification view of the close association between immune cells and a blood vessel forming a retinal-choroidal anastomosis in a retina with endothelial cell (EC)-specific loss of TGF-beta signaling.

## Increase in ICAM1 in retinal ECs and altered pericyte properties with loss of endothelial TGF-beta signaling

As noted in the Introduction, under inflammatory conditions, ECs secrete cytokines, express adhesion and co-stimulatory proteins to recruit and activate immune cells, and display immunogenic peptides bound to MHC proteins (*Pober and Sessa, 2014*; *Amersfoort et al., 2022*). Intercellular adhesion molecule 1 (ICAM1) is a general marker for vascular inflammation. Endothelial ICAM1 acts as a receptor for lymphocyte function-associated antigen 1 (LFA1), an integrin broadly expressed on leukocytes, and ICAM1-LFA1 binding promotes transmigration of leukocytes from blood to tissue (*Ding et al., 1999*). Immunostaining for ICAM1 in *Cdh5CreER;Tgfbr1^{CKO/-}*, *Fzd4^{-/-}*, and *Ndp^{KO}* retinas – each paired

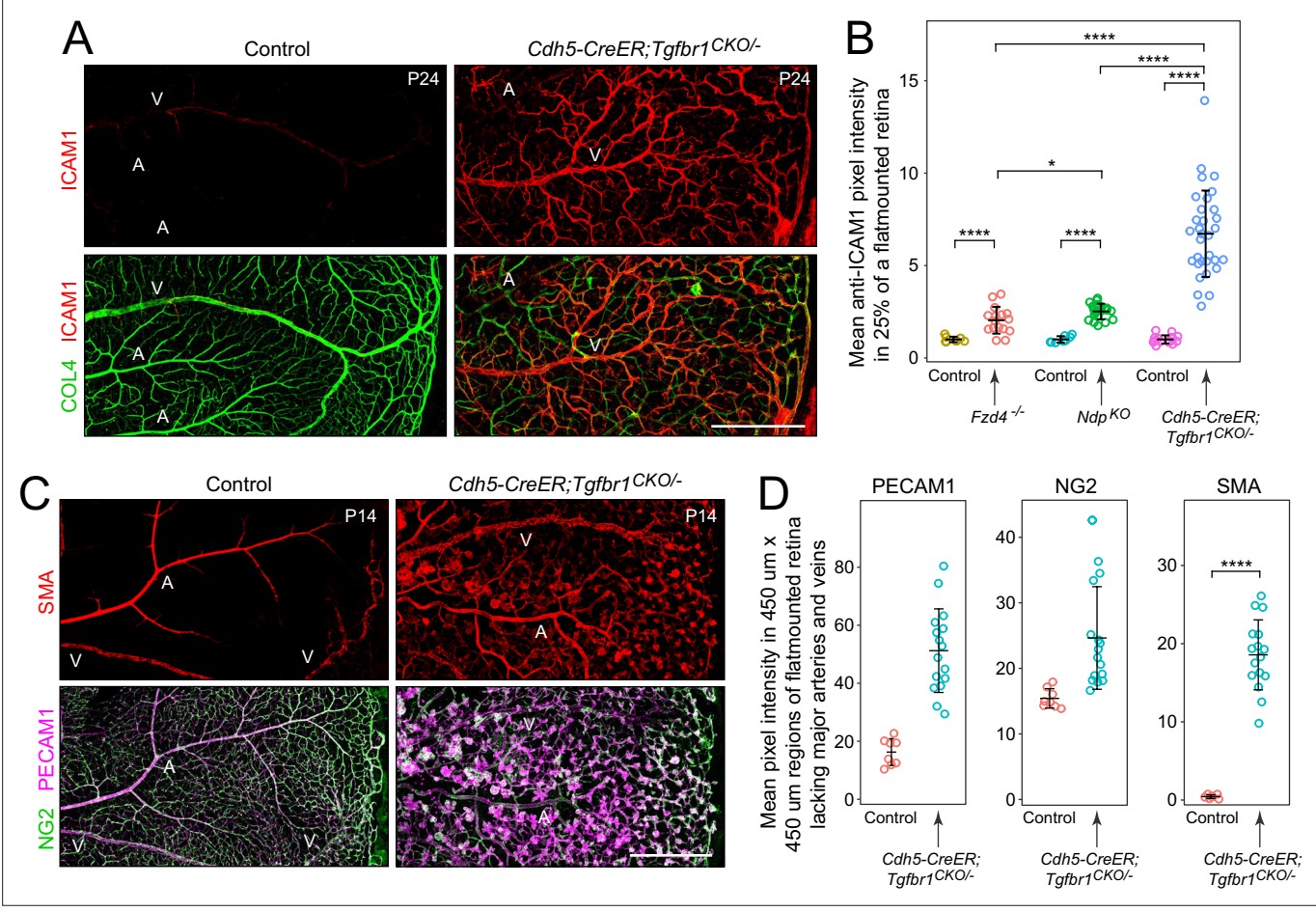

**Figure 7.** Intercellular adhesion molecule 1 (ICAM1) in endothelial cells (ECs) and smooth muscle actin (SMA) in pericytes in retinas with EC-specific loss of TGF-beta signaling. (**A**) ICAM1 in the retinal vasculature with EC-specific loss of TGF-beta signaling. Immunostaining conditions and image capture settings were identical across genotypes. (**B**) Quantification of the relative intensities of ICAM1 immunostaining in control vs. *Cdh5CreER;Tgfbr1^CKO/-*, control vs. *Fzd4^-/-* retinas, and control vs. *Ndp^KO* retinas. (**C**) Left, control retina flatmount showing strong SMA immunostaining in arteries, weak/patchy SMA immunostaining in veins, and undetectable SMA immunostaining in capillaries. Right, *Cdh5CreER;Tgfbr1^CKO/-* retina flatmount showing SMA immunostaining in all vessels, including vascular tufts. NG2 immunostaining (a pericyte marker) is shown in the images below. (**D**) Quantification of the relative intensities of PECAM1, NG2, and SMA immunostaining in flatmount control and *Cdh5CreER;Tgfbr1^CKO/-* retinas. Bars represent mean ± standard deviation, and p-values, calculated using the Wilcoxon rank-sum test, are shown as *<0.05, **<0.01, ***<0.001, and ****<0.0001. In (**A**) and (**C**), the retinal periphery is at the right. Scale bars in (**A**) and (**C**), 500 μm.

The online version of this article includes the following figure supplement(s) for figure 7:

**Figure supplement 1.** Examples of intercellular adhesion molecule 1 (ICAM1) immunostaining in individual experiments with control vs. *Cdh5CreER;Tgfbr1^CKO/-* retinas, control vs. *Fzd4^-/-* retinas, and control vs. *Ndp^KO* retinas used for the quantification shown in *Figure 7*.

with littermate control retinas processed in parallel – revealed a ~2-fold increase in endothelial ICAM1 in *Fzd4^-/-* and *Ndp^KO* retinas and a ~7-fold increase in *Cdh5CreER;Tgfbr1^CKO/-* retinas (*Figure 7A, B*, *Figure 7—figure supplement 1*). Control retina flatmounts show weak ICAM1 immunostaining in veins and undetectable ICAM1 in arteries and capillaries, whereas *Cdh5CreER;Tgfbr1^CKO/-* retina flatmounts show accumulation of ICAM1 in veins and in many small diameter vessels. The elevations in EC ICAM1 in these models of retinal hypovascularization correlate with the levels of retinal immune cells (*Figure 3*), suggesting that the level of vascular inflammation, and, more specifically, the level of ICAM1, is one determinant of the rate of egress of immune cells from blood to retina.

Changes in EC phenotype can lead to changes in the characteristics of pericytes, as seen, for example, with defects in platelet-derived growth factor signaling (*Lindahl and Betsholtz, 1998*). Although pericyte coverage appears to be minimally affected by loss of TGF-beta signaling (*Figure 1—figure supplement 3*), we observed a change in the relationship between NG2, a pericyte marker, and

smooth muscle actin (SMA), a smooth muscle marker, in flatmounts of *Cdh5CreER;Tgfbr1^CKO/-* retinas compared to control retinas. In control retinas, SMA immunostaining is intense on arteries, patchy on veins, and nearly undetectable on capillaries, with the latter staining strongly for NG2 (*Figure 7C*). In *Cdh5CreER;Tgfbr1^CKO/-* retinas, artery and vein SMA immunostaining is minimally affected, but capillaries and vascular tuft ECs immunostain for both NG2 and SMA (*Figure 7C and D*). These differences could reflect higher levels of SMA in *Cdh5CreER;Tgfbr1^CKO/-* pericytes or a change in actin polymerization state. The latter possibility is based on the observations of *Alarcon-Martinez et al., 2018*, that immunostaining for SMA in mouse retinal pericytes is strongly enhanced by actin polymerization.

## Transient defects in the BBB, altered pericyte properties, and localized immune infiltrates in the brain with loss of endothelial TGF-beta signaling

To determine whether the phenotypes associated with EC loss of TGF-beta signaling extend to CNS territories beyond the retina, we surveyed the brain in *Cdh5CreER;Tgfbr1^CKO/-* mice for leakage/transport of IgG into the parenchyma, pericyte changes (SMA immunostaining), vascular inflammation (ICAM1 immunostaining), and immune cell infiltration (*Figure 8*, *Figure 8—figure supplement 1*). There was no leakage into the parenchyma of Sulfo-NHS-biotin in the *Cdh5CreER;Tgfbr1^CKO/-* brain (data not shown). Interestingly, IgG accumulates in the *Cdh5CreER;Tgfbr1^CKO/-* brain parenchyma at P14 before resolving over the next week (*Figure 8A*), presumably reflecting enhanced IgG transport into or reduced IgG transport out of the brain (*Lafrance-Vanasse et al., 2025*). As in the retina, in *Cdh5CreER;Tgfbr1^CKO/-* brains, there was increased SMA immunoreactivity in capillary-associated pericytes (*Figure 8B*). In *Cdh5CreER;Tgfbr1^CKO/-* brains, there was scattered accumulation of immune cells and variably increased EC ICAM1 levels (*Figure 8—figure supplement 1*). These data suggest a pro-inflammatory state in ECs in young *Cdh5CreER;Tgfbr1^CKO/-* brains with an associated accumulation of IgG in the brain parenchyma.

After P30, mice with EC-specific loss of TGFBR1, TGFBR2, or both TGFBR1 and TGFBR2 exhibit an increase in the density of focal brain lesions that are characterized by bleeding and an accumulation of immune cells (*Figure 8C*). These brain lesions, which appear to be small cerebrovascular events (strokes), may explain the observation that otherwise healthy adult mice with EC-specific loss of TGF-beta signaling die at a rate of ~20% per month. Taken together, the histologic and mortality patterns suggest that mice with defects in TGF-beta signaling are prone to lethal cerebrovascular events.

## Altered brain endothelial gene expression with loss of endothelial TGF-beta signaling

To obtain an unbiased assessment of gene expression changes in brain ECs lacking TGF-beta signaling, vascular fragments from P14 control and *Cdh5CreER;Tgfbr1^CKO/-* brains were enriched by centrifugation through Ficoll and used as input for snRNAseq (*Figure 9*). The paucity of ECs in CNS tissue (several percent of cells) and the difficulty in freeing ECs from the extracellular matrix limit the yield of CNS ECs when using enzymatic dissociation methods and FACS sorting. By contrast, the Ficoll centrifugation protocol can rapidly enrich a large number of vascular fragments because it starts with a large mass of brain tissue (one or more mouse brains). We chose snRNAseq rather than single-cell (sc)RNAseq because snRNAseq has two advantages: (1) input vascular fragments can be frozen prior to nuclear isolation, and (2) the tissue is not subject to enzymatic dissociation, during which time changes in gene expression can occur (*Lacar et al., 2016*; *Hrvatin et al., 2018*).

snRNAseq on vascular fragments provides a snapshot of the transcriptomes of all of the major CNS cell classes, with EC nuclei enriched to ~15% of the total (*Figure 9A*). It is evident from visual inspection of the shapes and positions of cell clusters in the UMAP plots that gene expression in ECs undergoes a substantial change with EC-specific loss of TGF-beta signaling, whereas gene expression in other cell types undergoes far smaller changes. A volcano plot of transcriptome changes within the EC cluster shows several dozen transcripts with fold changes >4 and $-\log_{10}$ p-values greater than 50 (chosen as a conservative p-value cutoff for this comparison of 2475 control vs. 2944 mutant nuclei) (*Figure 9B*). *Figure 9C* shows UMAP plots for individual transcripts that exhibit increased abundance in mutant ECs (*Ahr*, *Nr5a2*, *Pcdh17*, and *Tgfbr3*), decreased abundance in mutant ECs (*Stra6*, *AU021092*, and *Htra3*), or little or no change in mutant ECs (*Icam1* and *Icam2*). In *Cdh5CreER;Tgfbr1^CKO/-* ECs, the elevation in *Tgfbr3* transcripts, which code for beta-glycan, an accessory (type

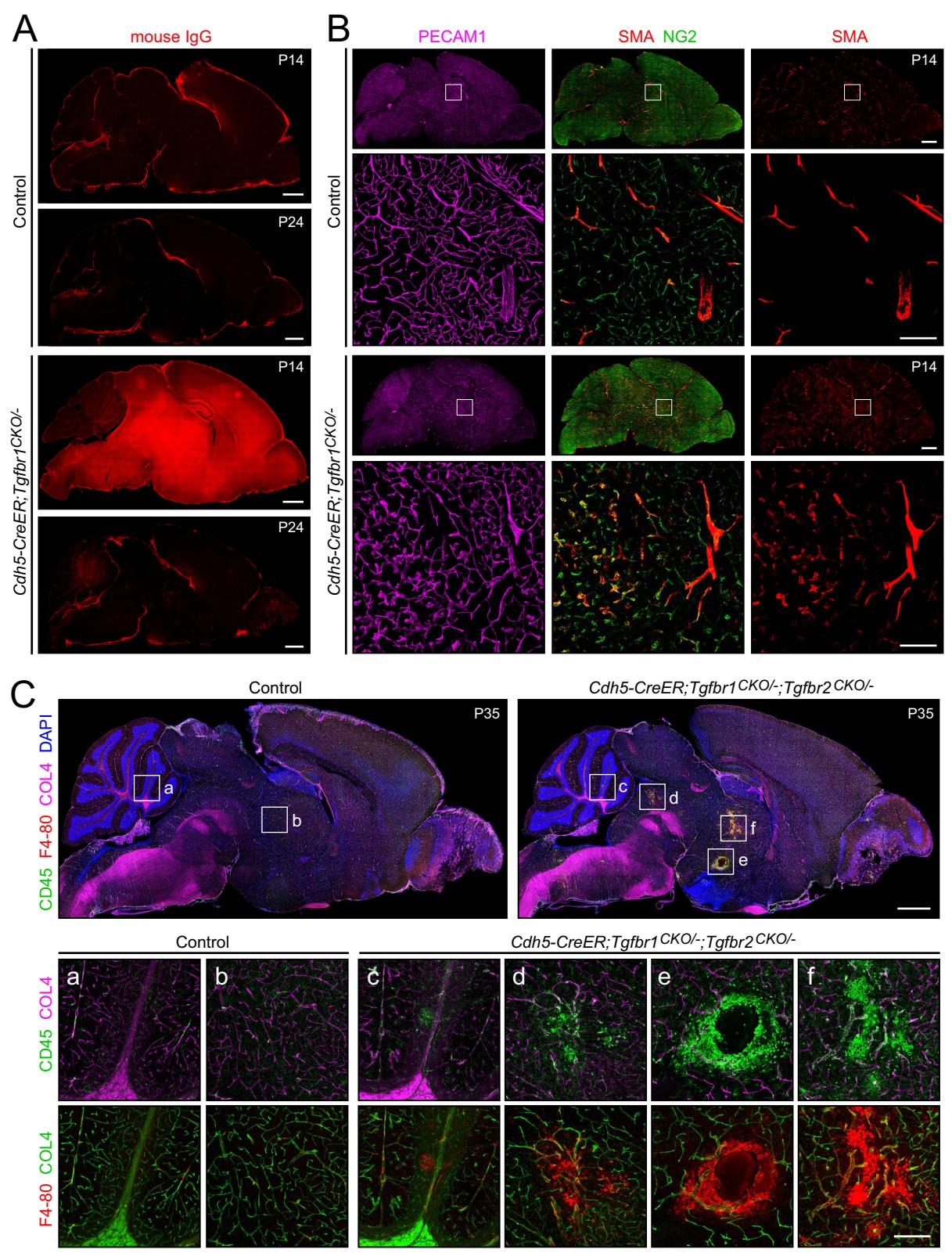

**Figure 8.** Transient IgG extravasation, smooth muscle actin (SMA) accumulation in pericytes, and immune cell infiltration in the brains of mice with endothelial cell (EC)-specific loss of TGF-beta signaling. (**A**) Endogenous IgG in control and *Cdh5CreER;Tgfbr1^CKO/-^* brains at postnatal day (P)14 and P24. IgG accumulation is minimal in control brains at P14 and P24 but is readily detectable in *Cdh5CreER;Tgfbr1^CKO/-^* brains at P14 but not at P24. (**B**) ECs (visualized with PECAM1) and pericytes (visualized with NG2) in control and *Cdh5CreER;Tgfbr1^CKO/-^* brains at P14. SMA immunostaining visualizes

*Figure 8 continued on next page*

*Figure 8 continued*

arterioles (continuous staining) and veins (patchy staining) in control and *Cdh5CreER;Tgfbr1^CKO/-* brains, and a subset of capillary-associated pericytes in *Cdh5CreER;Tgfbr1^CKO/-* brains. (**C**) Immune cells (CD45+ and F4-80+) in control and *Cdh5CreER;Tgfbr1^CKO/-* brains at P35. Control brains have minimal numbers of immune cells other than resident microglia. *Cdh5CreER;Tgfbr1^CKO/-* brains show localized regions with concentrated accumulations of immune cells. White squares marked with letters in the sagittal brain images (upper) are enlarged below. Scale bars, 1 mm for whole brain images and 200 μm for enlarged images.

The online version of this article includes the following figure supplement(s) for figure 8:

**Figure supplement 1.** Intercellular adhesion molecule 1 (ICAM1) immunostaining in *Cdh5CreER;Tgfbr1^CKO/-* brains.

III) TGFBR receptor, could represent a homeostatic response to reduced TGF-beta signaling since beta-glycan enhances the binding of TGF-beta ligands to the TGFBR1-TGFBR2 complex (*Heldin and Moustakas, 2016*). The lack of a significant change in *Icam1* transcript levels implies that the increase in ICAM1 immunostaining is a posttranslational phenomenon, either an increase in protein level or an increase in protein accessibility.

Gene set enrichment analysis (GSEA) confirms that the greatest changes in gene expression among CNS cell types in *Cdh5CreER;Tgfbr1^CKO/-* mice are in ECs, and it shows that EC transcriptome changes connected to the cell cycle and inflammation are the dominant categories (*Figure 9D*). The latter includes the categories 'interferon (IFN) gamma response', 'interferon (IFN) alpha response', 'IL2/STAT5 signaling', and 'inflammation'. An analysis of two categories of inflammation-associated proteins – components of the NF-kappa-B (NFkB) pathway and the integrins – shows (1) constitutive enrichment in CNS ECs of transcripts coding for the p50 subunit of NFkB (*Nfkb1*) and, most dramatically, NFkB inhibitor alpha (*Nfkbia*), and (2) with loss of TGF-beta signaling, upregulation of transcripts coding for multiple integrins, including alpha1, alpha2, alpha4, alpha6, and beta1 (*Figure 9—figure supplement 1A and B*). *Figure 9—figure supplement 1C* presents a pictorial summary of the transcripts coding for those chemokines, adhesion proteins, integrins, and TNF receptor superfamily members that are upregulated in CNS ECs in *Cdh5CreER;Tgfbr1^CKO/-* mice.

Immunostaining of whole-mount retinas for the p65 subunit of NFkB shows enrichment in both control and *Cdh5CreER;Tgfbr1^CKO/-* vasculature relative to nonvascular cells (*Figure 9—figure supplement 1D*). Immunostaining of whole-mount retinas for integrin alpha2 (ITGA2) shows undetectable staining in ECs in control vasculature and clear staining in *Cdh5CreER;Tgfbr1^CKO/-* vasculature. The change in ITGA2 immunostaining is in the same direction as the upregulation of *Itga2* transcripts in *Cdh5CreER;Tgfbr1^CKO/-* ECs (*Figure 9—figure supplement 1B*).

Immunostaining of whole-mount retinas for integrin alpha4 (ITGA4) shows enrichment in both control and *Cdh5CreER;Tgfbr1^CKO/-* vasculature relative to nonvascular cells (*Figure 9—figure supplement 1D*). Integrin alpha4/beta1 (very late antigen-4 [VLA-4]) is a heterodimer adhesion protein expressed on ECs and immune cells that is implicated in immune cell homing. It is also one of several integrins expressed by ECs that mediate cell-cell and cell-matrix interactions (*Guerrero and McCarty, 2018*; *Aman and Margadant, 2023*). In *Cdh5CreER;Tgfbr1^CKO/-* ECs, there is little change in vascular ITGA4 immunostaining intensity, but an increase in *Itga4* transcript levels (*Figure 9—figure supplement 1B and C*). These data suggest that there is an increase in both synthesis and degradation of ITGA4 in *Cdh5CreER;Tgfbr1^CKO/-* ECs.

The large number of transcriptional changes observed in control vs. *Cdh5CreER;Tgfbr1^CKO/-* CNS ECs imply substantial changes at the level of transcription and, by implication, the expression of transcription factors (TFs). Four TFs with the most dramatic increases in *Cdh5CreER;Tgfbr1^CKO/-* ECs are shown in the UMAP plots in *Figure 9C*, *Figure 9—figure supplement 2* the aryl hydrocarbon receptor (*Ahr*), a basic helix-loop-helix TF that reduces inflammatory responses in a variety of tissues and cell types (*Stockinger et al., 2024*); nuclear receptor subfamily 5 group A member 2 (*Nr5a2*), also referred to as liver receptor homologue-1, a TF with multiple roles in development and metabolism (*Fayard et al., 2004*); and two members of the thymocyte selection-associated high mobility group box gene family (*Tox* and *Tox3*), a TF family that is required for the development of innate immune cells and T-cells (*Aliahmad et al., 2012*). Immunostaining of retina flatmounts shows a several-fold increase in TOX levels in *Cdh5CreER;Tgfbr1^CKO/-* ECs compared to control ECs, with localization of the signal to the nucleus (*Figure 9—figure supplement 2B*).

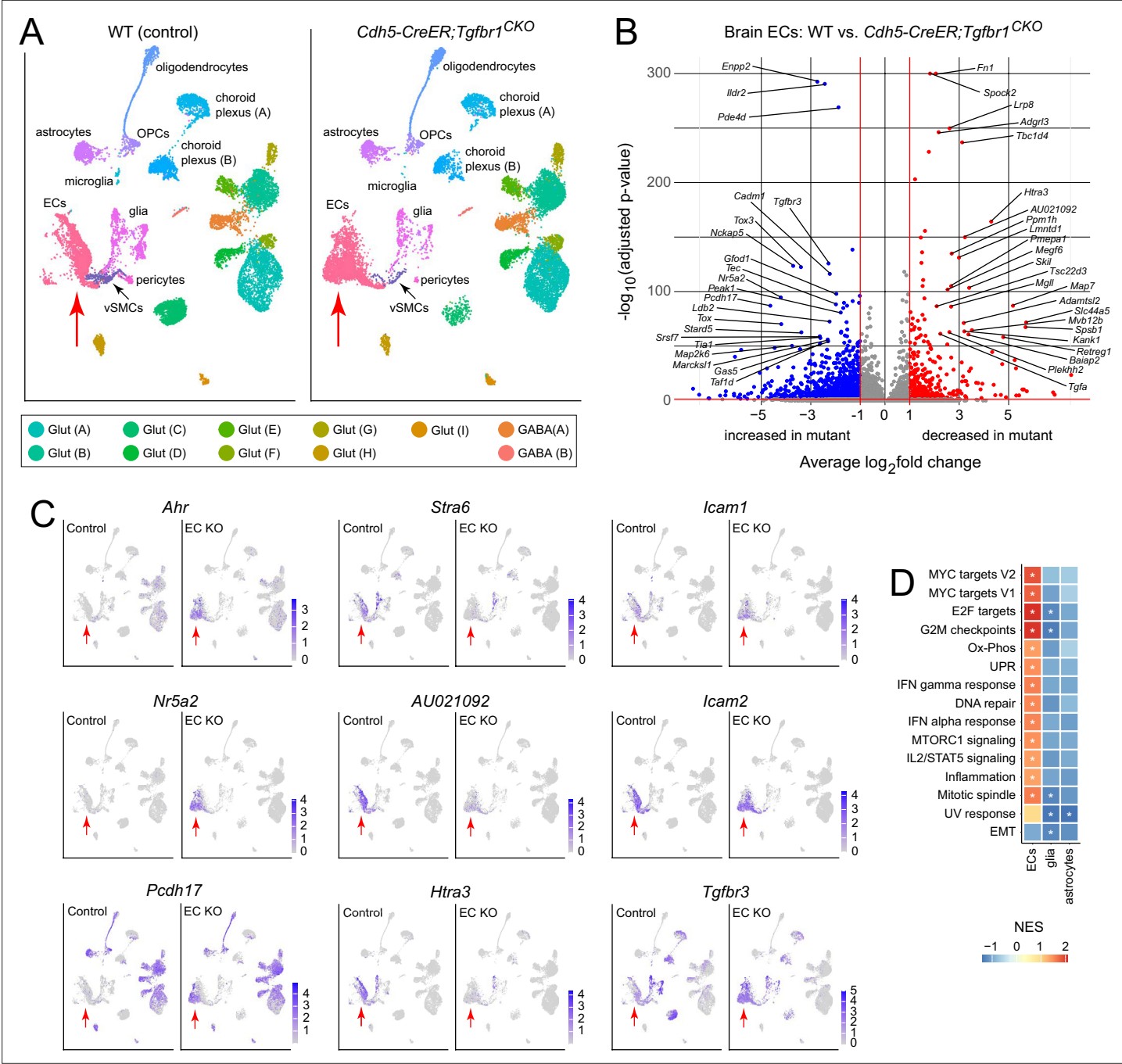

**Figure 9.** Single-nucleus RNAseq (snRNAseq) of control and *Cdh5CreER;Tgfbr1^CKO/-* vascular fragments enriched from the brain at postnatal day (P)14. (**A**) Uniform Manifold Approximation and Projection (UMAP) plots showing cell clusters encampassing the major cell types in the mouse brain, enriched for ECs, pericytes, and vascular smooth muscle cells (vSMCs). The locations of the EC clusters in the *Cdh5CreER;Tgfbr1^CKO/-* and control UMAP plots (vertical red arrows) are shifted, indicating substantial changes in their transcriptomes. Other cell cluster locations are largely unchanged. (**B**) Volcano plot showing transcripts from the EC cluster in control vs. *Cdh5CreER;Tgfbr1^CKO/-* snRNAseq. The labeled transcripts have adjusted $-\log_{10}$ p-values greater than 50. (**C**) UMAP plots as in (**A**) highlighting individual transcripts, with the EC cluster indicated by a vertical red arrow. Left column, UMAP plots for transcripts that are upregulated in *Cdh5CreER;Tgfbr1^CKO/-* ECs. Central column, UMAP plots for transcripts that are downregulated in *Cdh5CreER;Tgfbr1^CKO/-* ECs. Right column, *Icam1*, *Icam2*, and *Tgfbr3*. (**D**) Gene set enrichment analysis (GSEA) for the EC, glial, and astrocyte clusters in (**A**) showing the degree of enrichment in *Cdh5CreER;Tgfbr1^CKO/-* brains. The data used to generate the volcano plot in (**B**) and the GSEA in (**D**) are in **Supplementary file 1**. Additional analyses of differential expression are in **Supplementary file 2**. NES, normalized enrichment score. *, adjusted p-value<0.05.

The online version of this article includes the following figure supplement(s) for figure 9:

*Figure 9 continued on next page*

*Figure 9 continued*

**Figure supplement 1.** NF-kappaB, integrin, and other immune-related transcripts and proteins in retinal and brain endothelial cells (ECs) in control and *Cdh5CreER;Tgfbr1*<sup>CKO/-</sup> mice.

**Figure supplement 2.** Increased TOX levels in retinal endothelial cells (ECs) in *Cdh5CreER;Tgfbr1*<sup>CKO/-</sup> mice.

## Discussion

The experiments described here show that postnatal EC-specific loss of TGF-beta signaling in mice leads to aberrant angiogenesis in the retina and a pro-inflammatory state within the retina and brain. More specifically, EC-specific loss of TGF-beta signaling leads to: (1) reduced intraretinal vascularization, (2) CNV with occasional anastomoses connecting choroidal and intraretinal vasculatures, (3) infiltration of diverse immune cells into the retina, including macrophages, T-cells, B-cells, NK cells, and dendritic cells, (4) a close physical association between immune cells and vasculature, (5) a pro-inflammatory transcriptional state in CNS ECs, with increased ICAM1 immunoreactivity, and (6) increased SMA immunostaining in pericytes. A striking feature of the phenotypes studied here is their CNS specificity. Despite Cre-mediated recombination in ECs throughout the body directed by *Cdh5CreER*, the vascular and inflammatory phenotypes associated with EC-specific loss of TGF-beta signaling appear to be largely confined to the CNS.

Comparisons of the retinal phenotype with two other genetic models of retinal hypovascularization – loss of Norrin/Fzd4 signaling and loss of VEGF signaling – reveal interesting differences. While loss of either Norrin/Fzd4 or VEGF signaling leads to a near absence of intraretinal capillaries – similar to loss of TGF-beta signaling – the models differ in their immune phenotypes. The immune cell infiltrate is greatest with loss of TGF-beta signaling, more modest with loss of Norrin/Fzd4 signaling, and undetectable with loss of VEGF signaling. Interestingly, cleaved caspase 3 is abundant in immune cells with loss of TGF-beta signaling, but it is rare with loss of Norrin/Fzd4 signaling.

In considering the relationship between inflammation and vascular leakage with loss of TGF-beta signaling, it seems most likely that inflammation does not depend on vascular leakage in *Cdh5CreER;Tgfbr1*<sup>CKO/-</sup> retinas because Sulfo-NHS-biotin leakage into the retinal (and brain) parenchyma is minimal (*Figure 2*). By contrast, there could be a causal link between vascular leakage and inflammation with loss of Norrin/Fzd4 signaling because these retinas exhibit both leakage into the retinal parenchyma and inflammation (*Figures 2 and 3*). As noted in the Results section, CreER-mediated recombination in *Cdh5CreER;Tgfbr1*<sup>CKO/-</sup> mice after the completion of retinal vascular development (~P14) did not lead to immune cell infiltration in the retina, suggesting that the abnormal development/anatomy of the mutant retinal vasculature plays an essential role in the immune cell phenotype. This could reflect a role for the retinal hypoxia response in promoting a pro-inflammatory state, a hypothesis consistent with the absence of retinal inflammation with retina-specific loss of VEGF (*Figure 5*).

## A model for CNV

CNV is the hallmark feature of neovascular AMD. The original classification scheme for neovascularization in AMD posited two categories: occult, or type 1, neovessels were those located in the sub-RPE space, and classical, or type 2, vessels were those located in the subretinal space (i.e. between the RPE and retina) (*Gass, 1997*). More recently, a third type of neovessel has been recognized that consists of an anastomosis between choroidal and retinal vasculatures (*Freund et al., 2008*), and these type 3 vessels are observed in approximately one-third of eyes with newly diagnosed neovascular AMD (*Jung et al., 2014a*). Among patients with type 3 vessels in only one eye, the probability that the other eye will develop type 3 vessels within 3 years is close to 100% (*Gross et al., 2005*).

Multiple animal models of ocular neovascularization have been described (*Grossniklaus et al., 2010*; *Qiang et al., 2021*). In the simplest model of CNV, a focal laser-induced injury to the RPE and Bruch's membrane (the extracellular matrix that separates the RPE and the choroidal vessels) leads to the growth of choroidal vessels through the lesion (*Fabian-Jessing et al., 2022*). This response reveals the intrinsic angiogenic potential of the choroidal vasculature, which is normally held in check by Bruch's membrane and an intact RPE monolayer. Models of CNV that likely have greater relevance to pathogenic mechanisms in AMD have been developed in mice by altering – either singly or in combination – lipid/cholesterol metabolism, inflammation, and oxidative damage. These models include (1) KO of *Cyp27a1*, a ubiquitously expressed cytochrome P450 (*Omarova et al., 2012*), (2) *Ccr2/Ccl2*

double KO (*Takeda et al., 2009*), (3) ApoE overexpression combined with a high-fat diet (*Malek et al., 2005*), (4) *Ccl2/Cx3cr1* double KO combined with a low omega-3 polyunsaturated fatty acid diet (*Chan et al., 2008*), and (5) KO of superoxide dismutase-1 (*Sod1*) and aging for at least 1 year (*Imamura et al., 2006*). We note that CNV was observed in the *Ndp*^KO mice studied by *Beck et al., 2018*, although we did not observe CNV in our *Ndp*^KO mice.

Genetic models for pathologic angiogenesis that may be relevant to type 3 neovascularization include (1) overexpression of VEGF in rod photoreceptors (*Tobe et al., 1998*; *Ohno-Matsui et al., 2002*), (2) RPE-specific KO of the Von Hippel-Lindau gene (*Vhl*), which leads to activation of the hypoxia response and excessive production of VEGF (*Lange et al., 2012*), (3) rod photoreceptor-specific KO of *Vegfr1*, which codes for FLT-1, a decoy receptor that reduces VEGF signaling (*Luo et al., 2013*), and (4) KO of the very low-density lipoprotein receptor (*Vldlr*), which presumably perturbs lipid metabolism (*Heckenlively et al., 2003*; *Hu et al., 2008*).

Loss of TGF-beta signaling in ECs is mechanistically distinct from all of the above-mentioned models, and it has the added practical advantage that multiple CNV tufts are present in virtually every eye by 3 months of age without the need for dietary or other interventions (*Figure 1—figure supplement 1*).

## Inflammation in retinal and brain vascular disease

Local and/or systemic inflammatory markers, such as cytokines, are elevated in patients with AMD, diabetic retinopathy, and retinal vein occlusion (*Tang and Kern, 2011*; *Jung et al., 2014b*; *Kauppinen et al., 2016*). One characteristic of the retinal inflammatory state is increased leukocyte adhesion to retinal ECs, which is observed in patients with diabetic retinopathy and may be a causal factor in vaso-occlusive events (*Chibber et al., 2007*). In rats with streptozotocin-induced diabetes, ICAM1 and CD18 are elevated in retinal ECs, and genetic ablation of *Icam1* or *Cd18* or treatment with an ICAM1-blocking mAb reduces retinal leukostasis and vascular leakage (*Miyamoto et al., 1999*; *Joussen et al., 2004*). Neuroinflammation in general, and vascular neuroinflammation in particular, is also a feature of multiple non-retinal CNS diseases, including Alzheimer disease, stroke, and multiple sclerosis (*Ritson et al., 2024*).

Current evidence points to a combination of RPE oxidative damage, sub-RPE inflammatory cells, and complement activation as pathogenic mechanisms in neovascular AMD. In the early stages of AMD, inflammatory cells accumulate in the choroid and complement proteins accumulate in subretinal deposits, and, in the more advanced neovascular stage of AMD, choroidal neovascular tufts are associated with inflammatory cells (*Kauppinen et al., 2016*; *Armento et al., 2021*; *Heloterä and Kaarniranta, 2022*). Genetic evidence for a causal role of the complement system comes from the elevated AMD risk associated with the Y402H and I162V variants in the complement factor H gene, and less commonly with variants in the genes coding for complement factor 3 (C3), complement factor I (CFI), and the complement regulator SERPING1 (*Montezuma et al., 2007*).

In sum, the evidence presented here and in *Schlecht et al., 2017*, shows that loss of TGF-beta signaling in CNS ECs recapitulates some of the cardinal features of retinal and neurologic diseases associated with vascular inflammation. These experiments suggest that enhancing TGF-beta-dependent anti-inflammatory responses in ECs could represent a promising strategy for disease modulation (*Muniyandi et al., 2023*; *Hu et al., 2018*; *Hachana and Larrivée, 2022*; *Ravichandran and Heneka, 2024*).

## Materials and methods

**Key resources table**

| Reagent type (species) or resource | Designation | Source or reference | Identifiers | Additional information |
|---|---|---|---|---|
| Genetic reagent (*Mus musculus*) | *Cdh5CreER* | *Monvoisin et al., 2006* | Cdh5CreER | |
| Genetic reagent (*Mus musculus*) | *Tgfbr1*^CKO | *Larsson et al., 2001* | JAX 028701 | |

*Continued on next page*

*Continued*

| Reagent type (species) or resource | Designation | Source or reference | Identifiers | Additional information |
|---|---|---|---|---|
| Genetic reagent (*Mus musculus*) | *Tgfbr2^CKO^* | *Levéen et al., 2002* | JAX 012603 | |
| Genetic reagent (*Mus musculus*) | *Fzd4^KO^* | *Wang et al., 2001* | JAX 012823 | |
| Genetic reagent (*Mus musculus*) | *Ndp^KO^* | *Ye et al., 2009* | JAX 012287 | |
| Genetic reagent (*Mus musculus*) | *VEGF^CKO^* | *Gerber et al., 1999* | *VEGF^CKO^* | |
| Genetic reagent (*Mus musculus*) | *Vsx2-Cre* | *Rowan and Cepko, 2004* | *Vsx2-Cre* | |
| Genetic reagent (*Mus musculus*) | *Rosa26-LSL-SUN1-sfGFP* | *Mo et al., 2015* | JAX 030952 | |
| Genetic reagent (*Mus musculus*) | *Rosa26-LSL-tdTomato-2A-H2B-GFP* | *Wang et al., 2018* | JAX 030867 | |
| Antibody | Rat anti-mouse mAb PLVAP/MECA-32 | BD Biosciences | BD Biosciences # 553849 | (1:400) |
| Antibody | Rat anti-mouse mAb CD31 | BD Biosciences | BD Biosciences # 553370 | (1:400) |
| Antibody | Rat anti-mouse mAb ICAM-1 | Invitrogen | Invitrogen # 14-0542-82 | (1:400) |
| Antibody | Rat anti-mouse mAb F4/80 | Bio-Rad | Bio-Rad # MCA497G | (1:400) |
| Antibody | Rat anti-mouse mAb CD206 | Bio-Rad | Bio-Rad # MCA2235 | (1:400) |
| Antibody | Rat anti-mouse mAb PU.1/Spi-1 | R&D Systems | R&D Systems # MAB7124 | (1:400) |
| Antibody | Mouse mAb anti-alpha SMA, Cy3 conjugate | Sigma-Aldrich | Sigma-Aldrich # C6198 | (1:400) |
| Antibody | Mouse mAb anti-CLDN5, Alexa Fluor 488 conjugate | Thermo Fisher Scientific | Thermo Fisher Scientific # 352588 | (1:400) |
| Antibody | Mouse mAb anti-RPE65, Dylight 550 conjugate | Invitrogen | Invitrogen # MA5-16044 | (1:400) |
| Antibody | Rabbit polyclonal anti-Collagen IV | Novus Biologicals | Novus Biologicals # NB120-6586 | (1:400) |
| Antibody | Rabbit polyclonal anti-NG2 Chondroitin Sulfate Proteoglycan | Millipore | Millipore # AB5320 | (1:400) |
| Antibody | Rabbit mAb anti-ASC/TMS1 | Cell Signaling | Cell Signaling # 67824S | (1:400) |
| Antibody | Rabbit mAb anti-cleaved Caspase-3 | Cell Signaling | Cell Signaling # 9664S | (1:400) |
| Antibody | Rabbit mAb anti-HIF-1alpha | Cell Signaling | Cell Signaling # 36169S | (1:400) |
| Antibody | Rabbit mAb anti-P-SMAD1/5/9 | Cell Signaling | Cell Signaling # 13820S | (1:400) |
| Antibody | Armenian hamster mAb anti-CD3e | Invitrogen | Invitrogen # 14-0031-82 | (1:400) |
| Antibody | Goat polyclonal anti-CD45 | R&D Systems | R&D Systems # AF114 | (1:400) |
| Antibody | Goat polyclonal anti-Iba1 | Novus Biologicals | Novus Biologicals # NB100-1028 | (1:400) |

*Continued on next page*

*Continued*

| Reagent type (species) or resource | Designation | Source or reference | Identifiers | Additional information |
|---|---|---|---|---|
| Antibody | Chicken polyclonal anti-GFP | Aves Labs | Aves Labs # GFP-1020 | (1:400) |
| Antibody | Rabbit mAb anti-NF-κB p65. D14E12 | Cell Signaling Technology | Cell Signaling Technology # 8242S | (1:400) |
| Antibody | Rabbit mAb anti-Integrin alpha 2 (ITGA2); clone GEB | BosterBio | BosterBio # M01933 | (1:400) |
| Antibody | Rabbit mAb anti-Integrin alpha 4 (ITGA4); D2E1 | Cell Signaling Technology | Cell Signaling Technology # 8440 | (1:400) |
| Antibody | Rabbit mAb anti-TOX/TOX2; E6G5O | Cell Signaling Technology | Cell Signaling Technology # 36778S | (1:400) |

## Mice

The following mouse lines were used: *Cdh5CreER* (**Monvoisin et al., 2006**); *Tgfbr1^CKO^* (**Larsson et al., 2001**; JAX 028701); *Tgfbr2^CKO^* (**Levéen et al., 2002**; JAX 012603); *Fzd4^KO^* (**Wang et al., 2001**; JAX 012823); *Ndp^KO^* (**Ye et al., 2009**; JAX 012287); *VEGF^CKO^* (**Gerber et al., 1999**); *Vsx2-Cre* (**Rowan and Cepko, 2004**); *Rosa26-LSL-SUN1-sfGFP* (**Mo et al., 2015**; JAX 030952), and *Rosa26-LSL-tdTomato-2A-H2B-GFP* (**Wang et al., 2018**; JAX 030867). *Ndp* is located on the X-chromosome and therefore we refer to both female *Ndp^-/-^* and male *Ndp^-/Y^* mice as *Ndp^KO^*. All mice were housed and handled according to the approved Institutional Animal Care and Use Committee protocol of the Johns Hopkins Medical Institutions (protocol MO19M429).

## 4HT preparation and administration

4HT (Sigma-Aldrich H7904-25MG) was dissolved at 20 mg/ml in ethanol by extensive vortexing. Sunflower seed oil (Sigma-Aldrich S5007) was added to dilute the 4HT to 2 mg/ml, and aliquots were stored at –80°C. Thawed aliquots were mixed well before injections. All injections were performed intraperitoneally.

## Antibodies and other reagents

The following antibodies were used for tissue immunohistochemistry: rat mAb anti-mouse PLVAP/MECA-32 (BD Biosciences 553849); rat mAb anti-mouse CD31 (BD Biosciences 553370); rat anti-mouse ICAM-1 (Invitrogen 14-0542-82); rat mAb anti-mouse F4/80 (Bio-Rad MCA497G); rat mAb anti-mouse CD206 (Bio-Rad MCA2235); rat mAb anti-mouse PU.1/Spi-1 (R&D Systems MAB7124); mouse mAb anti-alpha SMA, Cy3 conjugate (Sigma-Aldrich C6198); mouse mAb anti-CLDN5, Alexa Fluor 488 conjugate (Thermo Fisher Scientific 352588); mouse mAb anti-RPE65, Dylight 550 conjugate (Invitrogen MA5-16044); rabbit polyclonal anti-Collagen IV (Novus Biologicals NB120-6586); rabbit polyclonal anti-NG2 Chondroitin Sulfate Proteoglycan (Millipore AB5320); rabbit mAb anti-ASC/TMS1 (Cell Signaling 67824S); rabbit mAb anti-cleaved Caspase-3 (Cell Signaling 9664S); rabbit mAb anti-HIF-1alpha (Cell Signaling 36169S); rabbit mAb anti-P-SMAD1/5/9 (Cell Signaling 13820S); Armenian hamster mAb anti-CD3e (Invitrogen 14-0031-82); goat polyclonal anti-mouse CD45 (R&D Systems AF114); goat polyclonal anti-Iba1 (Novus Biologicals NB100-1028); chicken polyclonal anti-GFP (Aves Labs GFP-1020); rabbit mAb anti-NFkappaB NF-κB p65 (D14E12; Cell Signaling Technology 8242S); rabbit mAb anti-Integrin alpha 2 (ITGA2; clone GEB, BosterBio M01933); rabbit mAb anti-Integrin alpha 4 (ITGA4; D2E1; Cell Signaling Technology 8440); rabbit mAb anti-TOX/TOX2 (E6G5O; Cell Signaling Technology 36778S). Alexa Fluor-labeled secondary antibodies and GS Lectin (Isolectin GS-IB4) were from Thermo Fisher Scientific. Alexa Fluor-labeled secondary goat anti-Armenian hamster IgG antibodies were from BioLegend. Texas Red Streptavidin was from Vector Laboratories (SA-5006). Sulfo-NHS-biotin was from Thermo Fisher Scientific (21217).

## Tissue processing and immunohistochemistry

Tissues were prepared and processed for immunohistochemical analysis as described by *Wang et al., 2012*, and *Zhou et al., 2014*. In brief, mice were deeply anesthetized with ketamine and xylazine and then perfused via the cardiac route with 1% paraformaldehyde (PFA) in phosphate-buffered saline (PBS). Non-ocular tissues were dissected and dehydrated in 100% cold methanol overnight at 4°C. Tissues were re-hydrated the following day in 1× PBS at 4°C for at least 3 hr before embedding in 3% agarose. Tissue sections of 100–200 μm thickness were cut using a Leica vibratome.

For whole-mount retinas, intact eyes were immersion fixed in 1% PFA in PBS at room temperature for 1 hr before the retinas were dissected. For eye sections, enucleated eyes were imbedded in optimal cutting temperature compound (Tissue-Tek 4853) and frozen in dry ice. Embedded eyes were cut into 14 μm sections with a Zeiss cryostat and stored on glass slides at –80°C. For immunostaining, sections were warmed to room temperature, fixed in 1% PFA at room temperature for 30 min, and washed in PBS before pre-blocking.

For vascular permeability analysis, mice were injected intraperitoneally with Sulfo-NHS-biotin (100–200 μl of 20 mg/ml Sulfo-NHS-biotin in PBS) 30–60 min prior to intracardiac perfusion. Covalently bound biotin was visualized in tissue sections or in whole-mount retinas with Texas Red-conjugated streptavidin.

Tissue sections or whole-mount retinas were permeabilized in PBSTC (1× PBS+1% Triton X-100+0.1 mM CaCl$_2$) overnight at 4°C, and subsequently incubated overnight at 4°C with primary antibodies, diluted in 1× PBSTC+7% normal goat or donkey serum. Samples were washed at least six times with 1× PBSTC over the course of 6–8 hr and subsequently incubated overnight at 4°C with secondary antibodies diluted in 1× PBSTC+7% normal goat or donkey serum. The next day, samples were washed at least six times with 1× PBSTC over the course of 6 hr and mounted in Fluoromount G (SouthernBiotech 0100-01).

## Epon embedding and processing

Following cardiac perfusion with 2% PFA and 2% glutaraldehyde in PBS, eyes were immersion fixed in the same fixative overnight at 4°C, treated for 90 min in osmium tetroxide on ice, dehydrated in an ethanol series, embedded in Epon, sectioned at 0.5 μm thickness, and stained with toluidine blue.

## Confocal microscopy

Confocal images were captured with a Zeiss LSM700 confocal microscope (20× or 40× objective) using Zen Black 2012 software, and processed with ImageJ, Adobe Photoshop, and Adobe Illustrator software. For experiments with control and mutant tissues, tissue processing, confocal imaging, and image processing were performed identically across genotypes unless stated otherwise.

## CNV quantification

For quantification of retinal CNV in frozen sections of whole eyes, 14 μm sections for analysis were spaced >200 μm apart so that each retina was sparsely sampled.

## Image analysis for cell counts

The density of immune cells was quantified from retina or choroid flatmount images by manually counting cells using the point-and-click 'Cell Counter' tool in Fiji-ImageJ (https://imagej.net/software/fiji/). From each quadrant of the retina, a 450 μm × 450 μm square was selected. For the choroid, the regions were 645 μm × 645 μm.

## Image analysis for immunostaining and Sulfo-NHS-biotin staining intensities

SMA, NG2, and PECAM1 immunostaining intensities were quantified from images of retina flatmounts that had been processed identically and imaged with identical confocal microscope settings. From each quadrant of the retina, a 450 μm × 450 μm square was selected that lacked large arteries or veins. Using the 'Analyze' tool and 'Mean gray value' function Fiji-ImageJ (https://imagej.net/software/fiji/), the mean intensity values were determined for the individual immunostaining channel. The same method was used for ICAM1 intensities, except the region analyzed consisted of an entire quadrant of retina. For each experiment, which included experimental and control retinas, the mean

pixel intensity in the ICAM1 channel was scaled for both experimental and control retinas so that the mean of the control value was set to 1.0. For quantifying streptavidin bound to Sulfo-NHS-biotin, small zones dispersed throughout the retina flatmount image and encompassing vessel-free parenchyma were captured as screenshots and their mean pixel intensities quantified with Fiji.

## Purification of brain vascular fragments

Vascular fragments were purified from P14 control and *Cdh5CreER;Tgfbr1*$^{CKO/-}$ mouse brains as described previously (*Hartz et al., 2018*), with some modifications. For each preparation, one mouse was anesthetized using isoflurane, surface sterilized with 70% ethanol, sacrificed by cervical dislocation, and its brain dissected and transferred to a 10 cm tissue culture dish. The brain was minced into small pieces (~1 × 1 mm) with a razor blade (100 strokes in each orthogonal direction) with two drops of Dulbecco's Phosphate Buffered Saline (DPBS; no calcium, no magnesium) added to maintain moisture (Gibco, 14190144).

The minced brain tissue was suspended in 10 ml of ice-cold DPBS supplemented with 5 mM EDTA (Invitrogen, AM9262) and 60 U/ml of RNasin-Plus RNase Inhibitor (Promega, N2615), and gently homogenized using a smooth pestle in a Thomas Pestle Tissue Grinder (3431E45) on ice with 5 strokes. The sample was incubated on ice for 10 min and then gently homogenized with 40 more strokes. To monitor the release of brain capillaries, 5 µl of the homogenate was briefly incubated at room temperature with DAPI and Isolectin GS-IB4 from *Griffonia simplicifolia* conjugated with Alexa Fluor 488 (Invitrogen, I21411) and then visualized under a fluorescent microscope.

The brain vascular fragments were pelleted by two rounds of centrifugation through 15% Ficoll PM 400 (Sigma-Aldrich, F4375). For the first round of centrifugation, 10 ml of the homogenate was vigorously mixed with 10 ml of 30% Ficoll PM 400. The sample was aliquoted into three 50 ml heavy-walled glass centrifuge tubes with round bottoms (Marienfeld, 3933041), and each tube was loaded with 6.7 ml of sample and then centrifuged at 5800 × *g* for 20 min at 4°C. The pellets, containing enriched vascular fragments, were saved. For the second round of centrifugation, the supernatant from the first round of centrifugation, which included a layer of myelin at its top, was transferred to a fresh 50 ml glass tube and vigorously mixed. The sample was aliquoted into three heavy-walled glass centrifuge tubes and centrifuged as described for the first round. The pellets from the two rounds of centrifugation were resuspended and pooled in a total volume of 10 ml of DPBS supplemented with 1% BSA and 40 U/ml of RNasin-Plus RNase Inhibitor ('DPBS+BR'). The sample was centrifuged in a heavy-walled glass centrifuge tube with a round bottom at 300 × *g* for 10 min at 4°C. The pellets were resuspended and pooled in a total volume of 1 ml of DPBS+BR by gently pipetting 10 times with a 1 ml pipette tip.

The 1 ml suspension was filtered through a Nylon Mesh Filter (Tisch Scientific; 300 µm mesh, ME17240) to remove large tissue fragments. The material trapped by the filter was rinsed one time with 1 ml of DPBS+BR. The resulting 2 ml suspension was then filtered through a PluriStrainer 20 µm filter (Cell Strainer, 43-50020-03) to capture vascular fragments. The PluriStrainer 20 µm filter was placed on top of a 50 ml tube and loaded in several steps with the suspension, which was allowed to drain by gravity flow. The vascular fragments retained on the filter were then washed once with 1 ml of DPBS+BR. The vascular fragments were recovered from the surface of the filter with several washes with 1 ml of DPBS+BR. The resulting 3 ml sample was centrifuged in a 12 ml heavy-walled glass centrifuge tube with a round bottom (Marienfeld, 3933011, 12 ml) at 300 × *g* for 10 min at 4°C. The pellet was resuspended in 500 µl of DPBS buffer with 40 U/ml of RNasin Plus RNase Inhibitor using a 1 ml Low Retention Pipette Tip (1000 µl Filtered Pipette Tips for Rainin LTS Pipette – RNase and DNase Free; 3840/CS, LTS-1000FT-CS). The suspended vascular fragments were placed in a 1.5 ml low-retention tube (Thermo Scientific, 3451) and centrifuged at 300 × *g* for 10 min at 4°C. The supernatant was removed, and the pellet was frozen on dry ice and stored at –80°C.

## snRNAseq

snRNAseq libraries were prepared using the PIPseq T20 3' Single Cell RNA Kit v4.0 PLUS or the PIPseq T10 Single Cell RNA Kit v5.0 (Fluent BioSiences; *Clark et al., 2023*; *Fontanez et al., 2024*). Frozen tissue was suspended in homogenization buffer (0.25 M sucrose, 25 mM KCl, 5 mM MgCl$_2$, 20 mM Tricine-KOH, pH 7.8) supplemented with 1 mM DTT, 0.15 mM spermine, 0.5 mM spermidine, EDTA-free protease inhibitor (Roche 11836 170 001), 0.5% IGEPAL-630, and 40 U/ml Protector RNase Inhibitor (Sigma, 03335402001). Samples were homogenized in a 2 ml Dounce homogenizer,

using 15 strokes with a loose-fitting pestle followed by 30 strokes with a tight-fitting pestle. The sample was filtered through a 10 μm filter (CellTrix, Sysmex, 04-004-2324), and nuclei were pelleted in low-retention 1.5 ml microcentrifuge tubes for 5 min at 500 × *g* at 4°C. Nuclei were washed twice with Nuclei Suspension Buffer (supplied in the Fluent BioSciences kit) supplemented with 1% BSA and 40 U/ml Protector RNase Inhibitor. Nuclei were counted, and ~40,000 nuclei (for the T20 kit) or ~20,000 nuclei (for the T10 kit) were used for library production following the Fluent BioSiences protocol. The resulting snRNAseq libraries were sequenced on an Illumina NovaSeq X Plus sequencer. Vascular fragments from two mouse brains were pooled for each snRNAseq library.

## Analysis of snRNAseq data

Reads were aligned with the PIPseeker program (Fluent BioSiences, version 3.3.0) using the pipseeker-gex-reference-GRCm39-2023.04 index provided by Fluent BioSiences. The CellBender program was used to detect empty droplets and remove background (version 0.3.2; *Fleming et al., 2023*). The SOLO program was used to remove doublets (scVI-tools 1.2.2-post2; *Bernstein et al., 2020*). Manual curation was used to further remove potentially mixed (doublet) nuclei, as well as clusters with multiple nuclei from samples AR7 and AR8. The data were normalized using a regularized negative binomial regression algorithm implemented with the SCTransform function (*Hafemeister and Satija, 2019*). Batch effects were corrected using the Seurat RPCAIntegration function. UMAP dimensional reduction was performed using the R uwot package (https://github.com/jlmelville/uwot, copy archived at *Melville, 2025*) integrated into the Seurat R package (*Melville, 2022*). Data for the various scatter plots were extracted using the Seurat AverageExpression function, and differential gene expression was analyzed using the Seurat FindMarkers function. The Wilcoxon rank-sum test was used to calculate p-values. The p-values were adjusted with a Bonferroni correction using all genes in the dataset. Data exploration, analysis, and plotting were performed using RStudio (*RStudio Team, 2020*), the tidyverse collection of R packages (*Wickham, 2017*), and ggplot2 (*Wickham, 2009*). Dotplots were generated with the default settings, including default normalization.

## GSEA

For GSEA (*Subramanian et al., 2005*), genes were ranked by the fold expression change between control and mutant datasets. The ranked gene list was used to detect enriched gene sets within the Broad Institute Hallmark Gene Sets using the fgsea R package (https://github.com/ctlab/fgsea; *Korotkevich et al., 2019*).

## Statistical analysis

All statistical values are presented as mean ± SD. The Wilcoxon rank-sum test was used to measure statistical significance. Statistical tests were carried out using the following web sites: https://www.socscistatistics.com/tests/signedranks/default2.aspx and https://www.omnicalculator.com/statistics/wilcoxon-rank-sum-test#how-do-i-calculate-wilcoxon-rank-sum-test. The statistical significance is represented graphically as n.s., not significant (i.e. $p > 0.05$); *, $p < 0.05$; **, $p < 0.01$; ***, $p < 0.001$; ****, $p < 0.0001$.

## Acknowledgements

Supported by the Howard Hughes Medical Institute. The authors thank David Mohr (Genetic Resources Core Facility, Johns Hopkins School of Medicine) for assistance with NextGen sequencing and Philip Seegren for helpful comments on the manuscript.

## Additional information

### Funding

| Funder | Grant reference number | Author |
|---|---|---|
| Howard Hughes Medical Institute | | Jeremy Nathans |

| Funder | Grant reference number | Author |
|--------|------------------------|--------|

The funders had no role in study design, data collection and interpretation, or the decision to submit the work for publication.

## Author contributions

Yanshu Wang, Conceptualization, Data curation, Formal analysis, Validation, Investigation, Methodology, Writing – original draft, Writing – review and editing; Amir Rattner, Data curation, Formal analysis, Validation, Investigation, Methodology, Writing – original draft, Writing – review and editing; Zhongming Li, Validation, Investigation, Methodology; Philip M Smallwood, Investigation, Methodology; Jeremy Nathans, Conceptualization, Formal analysis, Supervision, Funding acquisition, Investigation, Visualization, Methodology, Writing – original draft, Project administration, Writing – review and editing

## Author ORCIDs

Amir Rattner ⓘ https://orcid.org/0000-0001-9542-6212
Jeremy Nathans ⓘ https://orcid.org/0000-0001-8106-5460

## Ethics

All mice were housed and handled according to the approved Institutional Animal Care and Use Committee protocol of the Johns Hopkins Medical Institutions (protocol MO19M429).

Reviewer #1 (Public review): https://doi.org/10.7554/eLife.107018.3.sa1
Reviewer #2 (Public review): https://doi.org/10.7554/eLife.107018.3.sa2
Author response https://doi.org/10.7554/eLife.107018.3.sa3

# Additional files

## Supplementary files

Supplementary file 1. Differential gene expression data used to generate the volcano plot in *Figure 9B* and the gene set enrichment analysis (GSEA) plot in *Figure 9D*.

Supplementary file 2. Differential gene expression data used to generate the dotplots in *Figure 9— figure supplement 1A and B*.

MDAR checklist

## Data availability

The snRNAseq data have been deposited in the Gene Expression Omnibus (GEO) database (GSE306082).

The following dataset was generated:

| Author(s) | Year | Dataset title | Dataset URL | Database and Identifier |
|-----------|------|---------------|-------------|-------------------------|
| Wang Y, Rattner A, Li Z, Smallwood PM, Nathans J | 2025 | Vascular endothelial-specific loss of TGF-beta signaling as a model for choroidal neovascularization and central nervous system vascular inflammation | https://www.ncbi.nlm.nih.gov/geo/query/acc.cgi?acc=GSE306082 | NCBI Gene Expression Omnibus, GSE306082 |

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
