## [Editor Report · eLife Assessment]

Endothelial cell-specific loss of TGF-beta signaling in mice leads to CNS vascular defects, specifically impairing retinal development and promoting immune cell infiltration. The data are **solid**, showing that loss of TGF-beta signaling triggers vascular inflammation and attracts immune cells specific to CNS vasculature. These findings are **important**, highlighting TGF-beta's role in maintaining vascular-immune homeostasis and its therapeutic potential in neurovascular inflammatory diseases.

---

## [Referee Report · Reviewer #1 (Public review)]

Summary:

The manuscript analyses the effects of deleting the TgfbR1 and TgfbR2 receptors from endothelial cells at postnatal stages on vascular development and blood-retina barrier maturation in the retina. The authors find that deletion of these receptors affects vascular development in the retina but importantly it affects the infiltration of immune cells across the vessels in the retina. The findings demonstrate that Tgf-beta signaling through TgfbR1/R2 heterodimers regulates primarily the immune phenotypes of endothelial cells in addition to regulating vascular development, but has minor effects on the BRB maturation. The data provided by the authors provides a solid support for their conclusions.

Strengths:

(1) The manuscript uses a variety of elegant genetic studies in mice to analyze the role of TgfbR1 and TgfbR2 receptors in endothelial cells at postnatal stages of vascular development and blood-retina barrier maturation in the retina.

(2) The authors provide a nice comparison of the vascular phenotypes in endothelial-specific knockout of TgfbR1 and TgfbR2 in the retina (and to a lesser degree in the brain) with those from Npd KO mice (loss of Ndp/Fzd4 signaling) or loss of VEGF-A signaling to dissect the specific roles of Tgf-beta signaling for vascular development in the retina.

(3) The snRNAseq data of vessel segments from the brains of WT versus TgfbR1 -iECKO mice provides a nice analysis of pathways and transcripts that are regulated by Tgf-beta signaling in endothelial cells.

Weaknesses (Original Submission):

(1) The authors claim that choroidal neovascular tuft phenotypes are similar in TgfbrR1 KO and TgfbrR2 KO mice. However, the phenotypes look more severe in the TgfbrR1 KO rather than TgfbrR2 KO mice. Can the authors show a quantitative comparison of the number of choroidal neovascular tufts per whole eye cross-section in both genotypes?

(2) In the analysis of Sulfo-NHS-Biotin leakage in the retina to assess blood-retina barrier maturation, the authors claim that there is increased vascular leakage in the TgfbR1 KO mice. However, there does not seem like Sulfo-NHS-biotin is leaking outside the vessels. Therefore, it cannot be increased vascular permeability. Can the authors provide a detailed quantification of the leakage phenotype?

(3) The immune cell phenotyping by snRNAseq seems premature as the number of cells is very small. The authors should sort for CD45+ cells and perform single cell RNA sequencing.

(4) The analysis of BBB leakage phenotype in TgfbR1 KO mice needs to be more detailed and include some tracers in addition to serum IgG leakage.

(5) A previous study (Zarkada et al., 2021, Developmental Cell) showed that EC-deletion of Alk5 affects the D tip cells. The phenotypes of those mice look very similar to those shown for TgfbrR1 KO mice. Are D tip cells lost in these mutants by snRNAseq?

Comments on revisions:

The authors have addressed the major weaknesses that I raised with the original submission adequately in the revised manuscript.

---

## [Referee Report · Reviewer #2 (Public review)]

Summary:

The authors meticulously characterized EC-specific Tgfbr1, Tgfbr2, or double knockout in the retina, demonstrating through convincing immunostaining data that loss of TGF-β signaling disrupts retinal angiogenesis and choroidal neovascularization. Compared to other genetic models (Fzd4 KO, Ndp KO, VEGF KO), the Tgfbr1/2 KO retina exhibits the most severe immune cell infiltration. The authors proposed that TGF-β signaling loss triggers vascular inflammation, attracting immune cells - a phenotype specific to CNS vasculature, as non-CNS organs remain unaffected.

Strengths:

The immunostaining results presented are clear and robust. The authors performed well-controlled analyses against relevant mouse models. snRNA-seq corroborates immune cell leakage in the retina and vascular inflammation in the brain.

Comments on revisions:

The authors have revised the manuscript and addressed all my questions.

---

## [Author Response]

The following is the authors’ response to the original reviews.

**Reviewer #1 (Public review):**
Weaknesses:(1) The authors claim that choroidal neovascular tuft phenotypes are similar in TgfbrR1 KO and TgfbrR2 KO mice. However, the phenotypes look more severe in the TgfbrR1 KO rather than TgfbrR2 KO mice. Can the authors show a quantitative comparison of the number of choroidal neovascular tufts per whole eye cross-section in both genotypes?

Thank you for asking about this. Each VE-cad-CreER;TGFBR1 CKO/- and VE-cad-CreER;TGFBR2 CKO/- retina exhibits multiple zones of choroidal neovascularization. The examples in Figures 1 and Figure 1 – Figure supplements 1 and 2 are mostly from retinas with loss of TGFBR1, but we could have chosen similar examples from retinas with loss of TGFBR2. The quantification in the original version of Figure 1- Figure supplement 1 panel C had a labeling error. It actually showed the quantification choroidal neovascularization (CNV) in the sum of both VE-cad-CreER;TGFBR1 CKO/- and VE-cad-CreER;TGFBR2 CKO/- retinas, not only in VE-cad-CreER;TGFBR1 CKO/- retinas as originally labeled. The point that it made is that CNV is seen with loss of TGF-beta signaling but not in control retinas or retinas with loss of Norrin signaling. We have now updated that plot by separating the data points for VE-cad-CreER;TGFBR1 CKO/- and VE-cad-CreER;TGFBR2 CKO/- retinas, so that they can be compared to each other. The result shows ~2.5-fold more CNV in VE-cad-CreER;TGFBR2 CKO/- retinas compared to VE-cad-CreER;TGFBR1 CKO/-. We think it likely that a more extensive sampling would show little or no difference between these two genotypes – but the data is what it is. This is now described in the Results section.

We have also added a panel D to Figure 1- Figure supplement 1, which shows a retina flatmount analysis of CNV. This is done by mounting the retina with the photoreceptor side up so that the outer retina can be optimally imaged.

(2) In the analysis of Sulfo-NHS-Biotin leakage in the retina to assess blood-retina barrier maturation. The authors claim that there is increased vascular leakage in the TgfbR1 KO mice. However, it does not seem like Sulfo-NHS-biotin is leaking outside the vessels. Therefore, it cannot be increased vascular permeability. Can the authors provide a detailed quantification of the leakage phenotype?

Thank you for raising this point. Your comment prompted us to look at this question in greater depth with more experiments. We have expanded Figure 2 to show and quantify a comparison between control (i.e. phenotypically WT), NdpKO, and TGFBR1 endothelial KO and we have expanded the associated part of the Results section (Figure 2C and D). In a nutshell, control retinas show little Sulfo-NHS-biotin accumulation in or around the vasculature or in the parenchyma; NdpKO retinas show Sulfo-NHS-biotin accumulation in the vasculature and in the parenchyma (i.e., the area between the vessels); and VEcadCreER;Tgfbr1CKO/- retinas show Sulfo-NHS-biotin accumulation in the vascular tufts with minimal accumulation in the non-tuft vasculature and minimal leakage into the parenchyma. The conclusion is that the bulk of the retinal vasculature in TGFBR1 endothelial KO mice is minimally or not at all leaky – very different from the situation with loss of Norrin/Frizzled4 signaling.

(3) The immune cell phenotyping by snRNAseq is premature, as the number of cells is very small. The authors should sort for CD45+ cells and perform single-cell RNA sequencing.

Thank you for raising this point. For the revised manuscript, we have performed additional snRNAseq analyses using the same tissue processing protocol as for our original snRNAseq data. We have opted to homogenize the tissue and prepare nuclei (our original method) rather than dissociate the tissue and FACS sorting for CD45+ cells because the nuclear isolation approach is unbiased – we assume that nuclei from all cell types are present after tissue homogenization. By contrast, we cannot be certain that CD45 FACS will capture the full range of immune cells since some cells may not express CD45, may express CD45 at low level, or may be tightly adherent to other cells, such as vascular endothelial cell. Additionally, by following the original protocol, we can combine the original snRNAseq dataset and the new snRNAseq dataset. In the revised manuscript we present the snRNAseq data from the combination of the original and the more recent snRNAseq datasets (revised Figure 4; N=628 immune cell nuclei). The new analysis comes to the same conclusions as the original analysis: the immune cell infiltrate in the mutant retinas is composed of a wide variety of immune cells.

(4) The analysis of BBB leakage phenotype in TgfbR1 KO mice needs to be more detailed and include tracers as well as serum IgG leakage.

As described in our response to query 2, we have conducted additional experiments to look at vascular leakage in control, VE-cad-CreER;TGFBR1 CKO/-, and NdpKO retinas. We have also looked at Sulfo-NHS-biotin leakage in the VE-cadCreER;TGFBR1 CKO/- brain, and it is indistinguishable from WT controls. Since Sulfo-NHS-biotin is a low MW tracer (<1,000 kDa), this implies that loss of TGF-beta signaling does not increase non-specific diffusion of either low or high MW molecules. Therefore, the elevated levels of IgG in the brain parenchyma in young VE-cad-CreER;TGFBR1 CKO/- mice (Figure 8A) likely represents specific transport of IgG across the BBB. Such transport is known to occur via Fc receptors expressed on vascular endothelial cells, although it is normally greater in the brain-to-blood direction than in the blood-to-brain direction. For example, see Lafrance-Vanasse et al (2025) Leveraging neonatal Fc receptor (FcRn) to enhance antibody transport across the blood brain barrier. Nat Commun. 16:4143. This is now described in greater detail in the Results section.

(5) A previous study (Zarkada et al., 2021, Developmental Cell) showed that EC-deletion of Alk5 affects the D tip cells. The phenotypes of those mice look very similar to those shown for TgfbrR1 KO mice. Are D-tip cells lost in these mutants by snRNAseq?

Please note: Alk5 is another name for TGFBR1. This is noted in the second sentence of paragraph 4 of the Introduction. The reviewer is correct: there are a lot of similarities because these are exactly the same KO mice. Also, Zarkada and we used the same VEcadCreER to recombine the CKO allele. The proposed snRNAseq analysis would serve as an independent check on the diving (D) tip vs stalk cell analyses published in Zarkada et al (2021) Specialized endothelial tip cells guide neuroretina vascularization and blood-retina-barrier formation. Dev Cell 56:2237-2251. We have not gone in this direction because the question of tip vs. stalk cells and of subtypes of tip cells in WT vs. mutant retinas is beyond our focus on choroidal neovascularization and the role of immune cells and vascular inflammation. The proposed snRNAseq analysis would also require a major effort since tip cells are rare and must be harvested from large numbers of early postnatal retinas followed by FACS enrichment for vascular endothelial cells. Finally, we have no reason to doubt the results of Zarkada et al.

**Reviewer #2 (Public review):**
Summary:The authors meticulously characterized EC-specific Tgfbr1, Tgfbr2, or double knockout in the retina, demonstrating through convincing immunostaining data that loss of TGF-β signaling disrupts retinal angiogenesis and choroidal neovascularization. Compared to other genetic models (Fzd4 KO, Ndp KO, VEGF KO), the Tgfbr1/2 KO retina exhibits the most severe immune cell infiltration. The authors proposed that TGF-β signaling loss triggers vascular inflammation, attracting immune cells - a phenotype specific to CNS vasculature, as non-CNS organs remain unaffected.Strengths:The immunostaining results presented are clear and robust. The authors performed well-controlled analyses against relevant mouse models. snRNA-seq corroborates immune cell leakage in the retina and vascular inflammation in the brain.Weaknesses:The causal link between TGF-β loss, vascular inflammation, and immune infiltration remains unresolved. The authors' model posits that EC-specific TGF-β loss directly causes inflammation, which recruits immune cells. However, an alternative explanation is plausible: Tgfbr1/2 KO-induced developmental defects (e.g., leaky vessels) permit immune extravasation, subsequently triggering inflammation. The observations that vein-specific upregulation of ICAM1 staining and the lack of immune infiltration phenotypes in the non-CNS tissues support the alternative model. Late-stage induction of Tgfbr1/2 KO (avoiding developmental confounders) could clarify TGF-β's role in retinal angiogenesis versus anti-inflammation.

Thank you for raising this point. Your comment prompted us to look at this question in greater depth with more experiments. We have expanded Figure 2 to show and quantify a comparison between control (i.e. phenotypically WT), NdpKO, and TGFBR1 endothelial KO and we have expanded the associated part of the Results section (Figure 2C and D). In a nutshell, control retinas show little Sulfo-NHS-biotin accumulation in or around the vasculature or in the parenchyma; NdpKO retinas show Sulfo-NHS-biotin accumulation in the vasculature and in the parenchyma (i.e., the area between the vessels); and VEcadCreER;Tgfbr1CKO/- retinas show Sulfo-NHS-biotin accumulation in the vascular tufts with minimal accumulation in the non-tuft vasculature and minimal leakage into the parenchyma. The conclusion is that the bulk of the retinal vasculature in TGFBR1 endothelial KO mice is minimally or not at all leaky – very different from the situation with loss of Norrin/Frizzled4 signaling.

In the revised manuscript, we have expanded the Discussion section to address the two alternative hypotheses raised by the reviewer. Here are the relevant data in a nutshell: (1) vascular leakage into the parenchyma, as measured with sulfo-NHSbiotin, in TGFBR1 endothelial CKO retinas is far less than in NdpKO retinas, where nearly all ECs convert to a fenestration+ (PLVAP+) phenotype and there is leakage of sulfo-NHS-biotin, (2) ICAM1 in ECs in TGFBR1 endothelial CKO retinas increases several-fold more than in NdpKO or Frizzled4KO retinas, (3) TGFBR1 endothelial CKO retinas have more infiltrating immune cells than NdpKO or Frizzled4KO retinas, and (4) in TGFBR1 endothelial CKO retinas large numbers of immune cells are observed within and adjacent to blood vessels. We think that the simplest explanation for these data is that loss of TGFbeta signaling in ECs causes an endothelial inflammatory state with enhanced immune cell extravasation. That said, the case for this model is not water-tight, and there could be less direct mechanisms at play. In particular, this model does not explain why the inflammatory phenotype is limited to CNS (and especially retinal) vasculature.

Regarding the last sentence of the reviewer’s comment (“Late stage induction…”), we have tried activating CreER recombination at different ages and we observe a large reduction in the inflammatory phenotype when recombination is initiated after vascular development is complete. This observation suggests that the vascular developmental/anatomic defect – and perhaps the resulting retinal hypoxia response – is required for the inflammatory phenotype. In the revised manuscript we have expanded the Results and Discussion sections to describe this observation.

**Reviewer #1 (Recommendations for the authors):**
Suggestions for experiments:(1) The authors need to show a quantitative comparison of the number of choroidal neovascular tufts per whole eye crosssection in both genotypes (TgfbR1 and TgfbR2 KO mice).

Thank you for raising this point. The quantification in the original version of Figure 1- Figure supplement 1 panel C was mis-labeled. It quantifies choroidal neovascularization (CNV) in both VE-cad-CreER;TGFBR1 CKO/- and VE-cadCreER;TGFBR2 CKO/- retinas, not VE-cad-CreER;TGFBR1 CKO/- retinas only as originally labeled. The point it makes is that CNV is seen with loss of TGF-beta signaling but not in control retinas or retinas with loss of Norrin signaling. We have now corrected that plot by separating the data points for VE-cad-CreER;TGFBR1 CKO/- and VE-cad-CreER;TGFBR2 CKO/- retinas, so that they can be compared to each other. The result shows ~2.5-fold more CNV in VE-cad-CreER;TGFBR2 CKO/- retinas compared to VE-cad-CreER;TGFBR1 CKO/-. This is now described in the Results section.

(2) In the analysis of Sulfo-NHS-Biotin leakage in the retina to assess blood-retina barrier maturation. The authors should provide a detailed quantification of the leakage phenotype outside the vessels into the CNS parenchyma, both in the retina and brain, in TgfbR1 KO mice.

Thank you for raising this point. There is no detectable Sulfo-NHS-biotin leakage into the brain parenchyma in VE-cadCreER;TGFBR1 CKO/- mice. We have expanded Figure 2 to show and quantify the data for retinal vascular leakage (Figure 2C and D). The data show that in VE-cad-CreER;TGFBR1 CKO/- mice there is accumulation of Sulfo-NHS-biotin in the vascular tufts but minimal accumulation elsewhere in the retinal vasculature and minimal leakage of Sulfo-NHS-biotin into the retinal parenchyma.

(3) The immune cell phenotyping by snRNAseq is premature, as the number of cells is very small. The authors should sort for CD45+ cells and perform single-cell RNA sequencing to ascertain these preliminary data.

Thank you for raising this point. We have performed additional snRNAseq analyses using the same tissue processing protocol as for our original snRNAseq data to increase the numbers of cells. We have opted to homogenize the tissue and prepare nuclei (our original method) rather than dissociating the cells and FACS sorting for CD45+ cells because the nuclear isolation approach is unbiased – we assume that nuclei from all cell types are present. By contrast, we cannot be certain that CD45 FACS will capture the full range of immune cells, since some cells may not express CD45, may express CD45 at low level, or may be tightly adherent to other cells, such as vascular endothelial cell. Additionally, by following the original protocol, we can combine the original snRNAseq dataset of and the new snRNAseq dataset. In the revised manuscript we present the snRNAseq data from the combination of the original and the more recent snRNAseq datasets (revised Figure 4; N=628 immune cell nuclei). The new analysis comes to the same conclusion as in the original submission, namely that the immune cell infiltrate in the mutant retinas is composed of a wide variety of immune cells. The Results section has been expanded to describe this new data and analysis.

(4) The analysis of BBB leakage phenotype in TgfbR1 KO mice needs to be more detailed and include tracers as well as serum IgG leakage.

Sulfo-NHS biotin leakage in the VE-cad-CreER;TGFBR1 CKO/- brain is minimal, and it is indistinguishable from WT controls. Since Sulfo-NHS biotin is a low MW tracer (<1,000 kDa), this implies that loss of TGF-beta signaling does not increase non-specific diffusion of either low or high MW molecules. Therefore, the elevated levels of IgG in the brain parenchyma in young VE-cad-CreER;TGFBR1 CKO/- mice (Figure 8A) likely represents specific transport of IgG across the BBB. Such transport is known to occur via Fc receptors expressed on vascular endothelial cells, although it is normally greater in the brain-to-blood direction than in the blood-to-brain direction. For example, see Lafrance-Vanasse et al (2025) Leveraging neonatal Fc receptor (FcRn) to enhance antibody transport across the blood brain barrier. Nat Commun. 16:4143. This is now described in greater detail in the Results section.

(5) The authors should perform a more detailed RNAseq analysis of tip and stack (stalk) cells in TgfbrR1 KO mice to determine whether D tip cells are lost in these mutants by snRNAseq.

The proposed snRNAseq analysis would serve as an independent check on the diving (D) tip vs stalk cell analyses published by Zarkada et al, who analyzed the same VE-cad-CreER;TGFBR1 CKO/- mutant mice, although they refer to the TGFBR1 gene by its alternate name ALK5 [Zarkada et al (2021) Specialized endothelial tip cells guide neuroretina vascularization and blood-retina-barrier formation. Dev Cell 56:2237-2251]. We have not gone in this direction because the question of tip vs. stalk cells and of subtypes of tip cells in WT vs. mutant retinas is beyond our focus on choroidal neovascularization and the role of immune cells and vascular inflammation. The proposed snRNAseq analysis would also require a major effort since tip cells are rare and must be harvested from large numbers of early postnatal retinas followed by FACS enrichment for vascular endothelial cells.

Suggestions for improving the manuscript:(6) The statement that ECs acquire properties of immune cells (Page 2, Line 90) is incorrect. Endothelial cells may acquire characteristics of antigen presenting cells.

Thank you for that correction. Based on the review from Amersfoort et al (2022) (Amersfoort J, Eelen G, Carmeliet P. (2022) Immunomodulation by endothelial cells - partnering up with the immune system? Nat Rev Immunol 22:576-588) and the articles cited in it, we have changed the sentence to “Although vascular endothelial cells (ECs) are not generally considered to be part of the immune system, in some locations and under some conditions they acquire properties characteristic of immune cells, including secretion of cytokines, surface display of co-stimulatory or co-inhibitory receptors, and antigen presentation in association with MHC class II proteins (Pober and Sessa, 2014; Amersfoort et al., 2022).”

(7) The statement in Page 3, Line 100-101 [In CNS ECs, quiescence is maintained in part by the actions of astrocyte-derived Sonic Hedgehog, with the result that few immune cells other than resident microglia are found within the CNS (Alvarez et al., 2011).] is incomplete. Wnt signaling also suppresses the expression of leukocyte adhesion molecules from endothelial cells and therefore helps with immune cell quiescence.

Thank you for raising that point. We have expanded that sentence to include Wnt signaling in CNS endothelial cells, as described in the following reference: Lengfeld JE, Lutz SE, Smith JR, Diaconu C, Scott C, Kofman SB, Choi C, Walsh CM, Raine CS, Agalliu I, Agalliu D. (2017) Endothelial Wnt/beta-catenin signaling reduces immune cell infiltration in multiple sclerosis. Proc Natl Acad Sci USA 114:E1168-E1177.

(8) It may be beneficial for the reader to separate the results of the vascular phenotypes related to choroidal neovascularization compared to retinal vascular development.

Thank you for this suggestion. The two topics are partly overlapping: choroidal neovascularization is described in Figure 1, and retinal development is described in Figures 1 and 2. The challenge is that some of same images illustrate both phenotypes as in Figure 1, so the topics cannot be easily separated.

(9) In addition to comparing the phenotypes in Tgfb signaling mutant mice with Wnt signaling and VEGF-A signaling mutants, the authors should compare and contrast their data with those found in Alk5 KO mice, as there are a lot of similarities.

The reviewer has alerted us to a nomenclature challenge which we will try to resolve in the introduction: Alk5 is just another name for TGFBR1. The reviewer is correct: there are a lot of similarities between the present study and that of Zarkada et al (2021) because both use the same TGFBR1(=Alk5) CKO mice.

**Reviewer #2 (Recommendations for the authors):**
Figure 2For 2B, the authors should clarify whether the two regions shown in the Tgfbr1 KO retina (P14) represent central vs. peripheral areas, as phenotype severity varies.For 2C, does the uneven biotin accumulation reflect developmental gradients (e.g., central-peripheral maturation timing)?

Thank you for raising these points. Regarding Figure 2B, these images are all from the mid-peripheral retina, where the phenotype is moderately severe. This is now noted in the figure legend.

Regarding Figure 2C, the reviewer is correct that the pattern of Sulfo-NHS-biotin is uneven in VEcadCreER;Tgfbr1CKO/- retinas – it accumulates only in the tufts. We have expanded Figure 2C to show a comparison between control (i.e. phenotypically WT), NdpKO, and TGFBR1 endothelial KO retinas, and we have expanded the associated part of the Results section. In a nutshell, control retinas show little Sulfo-NHS-biotin accumulation in the vasculature or in the parenchyma; NdpKO retinas show Sulfo-NHS-biotin accumulation in the vasculature and in the parenchyma (i.e., the area between the vessels); and VEcadCreER;Tgfbr1CKO/- retinas show Sulfo-NHS-biotin accumulation in the vascular tufts with minimal accumulation in the non-tuft vasculature and minimal leakage into the parenchyma. The conclusion is that the bulk of the retinal vasculature in TGFBR1 endothelial KO mice is not leaky – very different from the situation with loss of Norrin/Frizzled4 signaling.

Figure 6The claim that PECAM1+ rings on veins reflect EC-immune cell binding is uncertain, as PECAM1 is also known to be expressed by immune cells. The complete correlation of PECAM1 and CD45 staining signals suggests that a subset of immune cells upregulates PECAM1. The VEcadCreER;Tgfbr1 flox/-; SUN1:GFP reporter would be helpful to delineate ECimmune cell proximity. Super-resolution imaging with Z-stacks could also resolve spatial relationships (luminal vs. abluminal immune cell adhesion).

Thank you for this comment. The reviewer is correct that, at the resolution of these images, we cannot determine whether the PECAM1 immunostaining signal is derived from ECs, from leukocytes, or from both. This is now stated in the Results section. The PECAM1-rich endothelial ring structure associated with leukocyte extravasation has been characterized in various publications, for example in (1) Carman CV, Springer TA. (2004) A transmigratory cup in leukocyte diapedesis both through individual vascular endothelial cells and between them. J Cell Biol 167:377-388 and (2) Mamdouh Z, Mikhailov A, Muller WA. (2009) Transcellular migration of leukocytes is mediated by the endothelial lateral border recycling compartment. J Exp Med 206:2795-2808. The ring structures visualized in Figure 6D by PECAM1 immunostaining conform to the ring structures described in these and other papers. In showing these structures, our point is simply that they likely represent sites of leukocyte extravasation. This is now clarified in the text. We have also added some additional references on leukocyte extravasation and the ring structures.

Figure 7A time-course analysis of ICAM1 would strengthen the mechanistic model. Does ICAM1 upregulation precede immune infiltration (supporting inflammation as the primary defect)? Given that immune cells appear by P14 (per snRNA-seq), is ICAM1 elevated earlier?

This is an interesting idea, but based on what is known about leukocyte adhesion and extravasation we predict that there will not be a clean temporal separation between ICAM1 induction and leukocyte adhesion/infiltration. That is, if the proinflammatory state causes an increase in the number of leukocytes, then as ICAM1 levels increase, leukocyte adhesion would also increase. Similarly, if the presence of leukocytes increases the pro-inflammatory state, then as the number of leukocytes increases, the levels of ICAM1 would be predicted to increase. Thus, we think that a time course analysis is unlikely to provide a definitive conclusion.

Figure 8-SF1In brain slices, a transient pan-IgG accumulation suggests a self-resolving defect in the BBB. However, this BBB impairment appears to be spatiotemporally distinct from ICAM1 upregulation. ICAM1 staining is restricted to the lesion site, aligning with immune cell-driven inflammation.

Thank you for raising these points. The reviewer is correct that these observations don’t fit together in a clear way. There does not appear to be a general increase in brain vascular permeability in VE-cad-CreER;TGFBR1 CKO/- mice, as shown by sulfo-NHS-biotin. However, there is a large and transient increase in IgG in the brain parenchyma, suggestive of a general vascular alteration, and – as the reviewer correctly notes – it is not accompanied by a generalized increase in ICAM1 vascular immunostaining. At this point, we don’t have any real insight into the mechanistic basis of the transient IgG increase.

Thank you for handling this manuscript.